# Continuous Temporal Graph Neural Networks

## Abstract

Temporal Graph Neural Networks (TGNNs) are pivotal in processing dynamic graphs. However, existing TGNNs primarily target one-time predictions for a given temporal span, whereas many practical applications require *continuous predictions*, that predictions are issued frequently over time. Directly adapting existing TGNNs to continuous-prediction scenarios introduces either significant computational overhead or prediction quality issues especially for large graphs. This paper revisits the challenge of *continuous predictions* in TGNNs, and introduces CODEN, a TGNN model designed for efficient and effective learning on dynamic graphs. CODEN innovatively overcomes the key complexity bottleneck in existing TGNNs while preserving comparable predictive accuracy. Moreover, we further provide theoretical analyses that substantiate the effectiveness and efficiency of CODEN, and clarify both its relationship to and improvements over attention-based models. Our evaluations across five dynamic datasets show that CODEN surpasses existing performance benchmarks in both efficiency and effectiveness, establishing it as a superior solution for continuous prediction in evolving graph environments.

## 1 Introduction

Temporal Graph Neural Networks (TGNNs) are prominent approaches for capturing the temporal information in the evolving process of graph-structured data and have delivered remarkable performance in recent years. TGNNs have been applied in many modern services built on temporal graphs, such as the social networks with frequent user interactions (Mislove et al., 2007; Zhao et al., 2016), financial networks with market fluctuations (Song et al., 2023; Xu et al., 2023), and communication networks with varying traffic loads (Shao et al., 2022b;a).

Despite the success of TGNNs, most existing TGNNs focus on one-time prediction at a single time step (e.g., the final snapshot). However, *continuous queries* are essential in many applications. In e-commerce, for example, the historical interactions between users and goods are used to predict the gross merchandise value, which directly impacts the user benefits (Ye et al., 2022). As users continuously search on the trading platform (e.g., Amazon, Alibaba Taobao) to retrieve prediction results for informed decision-making, it is crucial to timely refresh the historical interactions so as to make accurate predictions for these queries. Moreover, many social platforms such as LinkedIn experience millions of interactions between users per day (Borisyuk et al., 2024), where the rapidly evolving social relationships demand continuously refreshed predicted recommendation results. In such scenarios, however, existing studies on TGNNs, originally designed for one-time prediction, become inefficient for continuous prediction especially on large graphs (Feng et al., 2024; Skarding et al., 2021).

**Limitations of existing works.** To support the continuous prediction in TGNNs, there are three available categories of approaches, namely, *single-snapshot* GNN methods, *RNN-based* methods and *attention-based* methods. The *single-snapshot* methods (Zheng et al., 2022; Guo et al., 2022) simply utilize the embeddings of the current graph to answer the prediction request from users. To achieve high efficiency for continuous predictions, single-snapshot GNNs neglect the knowledge learned in previous snapshots while learning new patterns, resulting in degraded model performance in evolving scenarios. The *RNN-based* methods (Pareja et al., 2020; Panagopoulos et al., 2021) rely on an iterative framework, where the node states are updated sequentially at each time step based on the current input and the previous state. Due to the inherent *forgetting* mechanism of RNNs (Graves & Graves, 2012), these methods have a limited memory horizon and struggle to effectively model long-term interactions. The *attention-based* methods (Sankar et al., 2020; Guo et al., 2019) employ the commonly used attention mechanism (Vaswani et al., 2017) to compute the attention score between previous node embeddings with the current hidden state and selectively retain the significant part for predictions. However, as historical interactions accumulate, attention-based methods will

Table 1: Comparison of different categories in TGNNs. $K$ stands for the number of convolution layers, $p$ means the number edge updates, $\lambda$ is the embedding distance parameter, and $F$ is the dimension of node features. We set the graph propagation error $\epsilon$ as $O(\frac{1}{n^{(t)}})$ and $x_{max}$ is defined as: $\{x_{max}\}_i = \max_{1 \leq j \leq F} |X_{ij}^{(t)}|$. The references and details of these methods can be found in the Appendix A.7.

| Categories | Methods | Minimal Complexity for Updating $p$ Edges | | Efficiency | Memorization |
| --- | --- | --- | --- | --- | --- |
| | | Update Embeddings | Update States | | |
| Single-snapshot | Instant, DynAnom, IDOL | $O\left(p\sum_{i=1}^F \|x_i^{(t)}\|_1\right)$ | N.A. | | |
| RNN-based | TGN, TGCN, EvolveGCN, MPNN, ROLAND, TGL | $O(Km^{(t)}F)$ | $O(pn^{(t)}F^2)$ | | |
| Attention-based | DySat, TGAT, ASTGCN, DNNTSP, SEIGN, DyGFormer | $O(Km^{(t)}F)$ | $O\left(p\left(n^{(t)}\right)^2 F\right)$ | | |
| Ours | CODEN | $O\left(p\sum_{i=1}^F \|x_i^{(t)}\|_1\right)$ | $O\left(\frac{\|x_{max}\|_1+n^{(t)}}{\lambda}pF^2\right)$ | | |

practically lead to noticeable time consumption that grows quadratically with $T$[1], which degrades prediction efficiency over extended time spans. We summarize the characteristics and the complexity of each paradigm in Tab. 1. Given the imbalanced trade-off between accuracy and efficiency of existing methods, a natural question arises: *given limited computation resources, is there a model that can integrate the advantages of existing paradigms and take into account both accuracy and efficiency in evolving scenarios?*

These questions motivate us to design CODEN, a scalable framework targeting the continuous prediction on TGNNs. When handling frequent updates in dynamic scenarios, CODEN distinguishes itself among mainstream TGNNs by balancing both effectiveness and efficiency. On the theoretical side, CODEN achieves a complexity for updating node states that matches the minimal complexity of leading TGNNs, as demonstrated in Tab. 1. Furthermore, we establish a theoretical connection between CODEN and the widely-used attention mechanism, providing a strong foundation for its effectiveness. A set of derived theories further unveil the rationale behind CODEN's superior efficiency and predictive performance. Compared with the state-of-the-art method TGL (Zhou et al., 2022) which use 98.3 hours to finish the continuous prediction on billion-scale graph *Papers100M* (111M nodes, 1.6B edges) averagely, our method CODEN finishes in 3.3 hours on a single GPU.

**Contributions.** This paper introduces CODEN, an efficient TGNN for continuous prediction that maintains node states via efficient incremental updates on evolving graphs. We prove that CODEN achieves principled compression of historical interactions, yielding a superior accuracy–efficiency trade-off compared to state-of-the-art methods. Theoretically, we establish a connection between our temporal-processing paradigm and kernel attention mechanism, showing that the compression effect can be interpreted as masking redundant historical information. Additionally, we propose a matched ablation variant based on the attention mechanism to further highlight the efficiency and representation quality of CODEN. Across diverse real-world datasets including large-scale graphs, CODEN delivers up to $44.80\times$ faster training while matching or surpassing strong baselines in predictive accuracy.

## 2 PRELIMINARY AND RELATED WORK

### 2.1 NOTATIONS

Consider a directed and attributed graph at time $t$ ($t \geq 0$), denoted as $\mathcal{G}^{(t)} = (\mathcal{V}^{(t)}, \mathcal{E}^{(t)}, X^{(t)})$. Here $\mathcal{V}^{(t)} = \{v_1, v_2, ..., v_{n^{(t)}}\}$ represents the set of $n^{(t)}$ nodes, $\mathcal{E}^{(t)}$ constitutes the set of $m^{(t)}$ edges and $X^{(t)} = \{x_1^{(t)}, x_2^{(t)}, ..., x_F^{(t)}\}$ is the set of node attribute matrix with $x_i^{(t)} \in \mathbb{R}^{n^{(t)} \times 1}$ representing the attribute vector in $i$-th dimension. Here we denote $A^{(t)}$ and $D^{(t)}$ as the adjacency matrix and the degree matrix at time $t$ respectively. For each node $v \in \mathcal{V}^{(t)}$, $\mathcal{N}_{out}^{(t)}(v)$ denotes the out-neighbors of $v$ and $\mathcal{N}_{in}^{(t)}(v)$ denotes the in-neighbors at time $t$. In the setting of a continuous-time dynamic graph (CTDG) (Zheng et al., 2022; Li et al., 2023b), the graph is treated as an initial graph $\mathcal{G}^{(0)} = (\mathcal{V}^{(0)}, \mathcal{E}^{(0)})$ and the subsequent update events consisting of edge insertions and deletions. [2]

---

[1]In TGNNs, $T$ may scale up to the number of edges.

[2]The insertion and deletion of vertices can be managed by adding or removing their respective incident edges. Therefore, this paper focuses exclusively on edge updates and we have $n^{(t)} = n$.

We denote the set of update events as $\Gamma = \{e_1, e_2, ..., e_t, ...\}$, where the edge update $e_t = \{u_t, v_t, t\}$ indicates that the link between nodes $u_t$ and $v_t$ will be toggled at time $t$—added if absent, or deleted if present—transforming graph $\mathcal{G}^{(t-1)}$ into $\mathcal{G}^{(t)}$. We leave the notation table in Appendix A.1.

**Problem Definition.** Given batches of update events $\{\Gamma_1, \Gamma_2, ..., \Gamma_k, ...\}$, where each $\Gamma_k$ ends at time $t_k$, our objective is to efficiently learn the node state $\boldsymbol{M}^{(t)}$ for $t \in T = \{t_1, t_2, ..., t_k, ...\}$ and continuously conduct the downsteam tasks with the current graph $\mathcal{G}^{(t)}$. The state $\boldsymbol{M}^{(t)}$ should not only represent the current structure of the graph at time $t$, but also capture the historical evolution of each node by integrating dynamic changes over time. This evolving state is crucial for effectively encoding long-term temporal dependencies and can be leveraged in downstream tasks such as temporal node classification. Our approach aims to achieve this with limited computational resources, ensuring efficiency as the large-scale graph continuously evolves.

## 2.2 GENERAL MESSAGE PASSING WITH CTDG

In temporal graphs, the neighbors of a node can vary with time. Such evolvement must be captured in the learning process for accuracy. A common approach (Zhou et al., 2022; Rossi et al., 2020) is to maintain the node states $\boldsymbol{M}^{(t)} \in \mathbb{R}^{n^{(t)} \times F'}$ by summarizing the historical information of neighbors until timestep $t$, where $F'$ is the dimension of the state. Specifically, given a new edge connecting $e_{t+1} = \{u, v, t+1\}$ from node $u$ to node $v$, these algorithms update the state of node $u$ as:

$$\boldsymbol{M}^{(t+1)}(u) = \text{UPDT}\left(\boldsymbol{M}^{(t)}(u), \text{MSG}\left(\mathcal{G}^{(t)}, \boldsymbol{M}^{(t)}(u), e_{t+1}\right)\right), \quad (1)$$

where $\text{MSG}(\cdot)$ is the message functions to aggregate the new neighborhood information triggered by update $e_{t+1}$ and $\text{UPDT}(\cdot)$ represents the temporal-processing units to summarize the new evolution information. Under this framework, the states of node $v$ and its neighboring nodes $u$ and $v$ will also be updated. Existing categories of TGNNs are primarily differentiated by how they establish dependencies among time steps using $\text{MSG}(\cdot)$ and $\text{UPDT}(\cdot)$ functions. Each paradigm offers its unique trade-off between accuracy and efficiency, which is demonstrated in Tab. 1.

## 2.3 MAINSTREAM TGNNs

**Single-snapshot Methods.** The single-snapshot methods aim to perform efficient update with the minimal time cost based on the previous graph and new interactions. Specifically, Instant (Zheng et al., 2022), DynAnom (Guo et al., 2022) and IDOL (Zhu et al., 2024) explore the invariant rules of graph propagation and conduct an invariant-based algorithm of Personalized PageRank (PPR) to refresh the node embedding locally. However, only focusing on the current snapshot will miss key interactions of past time steps, leading to sub-optimal prediction results.

**RNN-based Methods.** RNN-based methods are generally based on the classical RNN (Cho et al., 2014), GRU (Chung et al., 2014), LSTM (Hochreiter & Schmidhuber, 1997) algorithm, etc., which simply use both the current input and the previous hidden state to iteratively capture temporal dependencies. For example, TGCN (Zhao et al., 2019) and EvolveGCN (Pareja et al., 2020) incorporate the Graph Convolutional Network (GCN)(Kipf & Welling, 2017) in the $\text{MSG}(\cdot)$ function to conduct graph propagation while coupling with an RNN module to learn node states. Similarly, TGL (Zhou et al., 2022) introduces a parallel framework leveraging multiple GPUs and extends various RNN-based methods to large-scale graphs. However, this framework is highly resource-intensive and demonstrates limited efficiency when deployed on limited resources. Furthermore, the simplistic recurrence mechanism of RNN often leads to difficulties in retaining historical information, especially as the graph size and temporal scope expand (Li et al., 2024a; Gers et al., 2000).

**Attention-based Methods.** To address the forgetting problem of RNNs, attention-based methods rely on the attention mechanism (Vaswani et al., 2017) and encodes the position of sequences, enabling the information flow from past to current states. For example, DySat (Sankar et al., 2020) employs the generalized GAT module (Veličković et al., 2018) to integrate the states of each node from different time steps to generate refreshed ones. ASTGCN (Guo et al., 2019) and DNNTSP (Yu et al., 2020) incorporate the attention mechanism to capture the spatial and temporal dependency for enhanced state quality. While attention mechanisms can retain the most relevant parts of the sequences to avoid the forgetting issue, they can become computationally expensive when processing with frequent updates (Thomas et al., 2020).

## 3 METHODOLOGY

The goal of CODEN is to minimize the cost of learning the new node state $M^{(t)}$ across a series of time steps. The realization of this goal relies on two key components: (a) an efficient routine that aggregates newly arrived neighborhood information (reflected in MSG($\cdot$) of Equ. 1); and (b) a lightweight paradigm to enable the summarization of evolution information (reflected in UPDT($\cdot$)). For component (a), state-of-the-art methods such as DyGFormer (Yu et al., 2023b), EvolveGCN (Pareja et al., 2020), and ROLAND (You et al., 2022) rely heavily on a large number of trainable parameters for neighborhood aggregation. Frequent updates necessitate continuous re-training of these parameters, resulting in significant computational overhead. For component (b), dynamic scenarios require the node states to continuously summarize the evolving information. However, as highlighted in Tab. 1, existing methods often struggle to balance accuracy and efficiency, leading to suboptimal performance. To address these challenges, we first decouple graph propagation from the training process and incrementally integrate permutation information from updates (Sec. 3.1). Then we perform a batch processing based on the upper bound of disruptions from updates (Sec. 3.2), where the theoretical analysis shows that the refreshed node states effectively compress historical graph information with an approximation guarantee. Finally, we present a theoretical analysis explaining CODEN's effectiveness and efficiency (Sec. 3.3), including its relationship to and improvements over attention-based models. All missing proof of lemmas and propositions can be found in the Appendix.

### 3.1 INCREMENTAL STATE UPDATE

**Node Embeddings.** Existing RNN-based and attention-based methods are inclined to adopt the classical learnable encoder (e.g., GCN (Kipf & Welling, 2017), GraphSage (Hamilton et al., 2017), GAT (Veličković et al., 2018)) to generate node embeddings, which require costly retraining whenever the graph changes. To address this issue, we adopt the Personalized PageRank (PPR) based embedding, because it is more scalable in computation (Liao et al., 2022; Wang et al., 2021a; Zhu et al., 2024) and it can also alleviate over-smoothing issues (Wu et al., 2022). Specifically, after setting $\boldsymbol{P}^{(t)} = \boldsymbol{A}^{(t)^\top} \boldsymbol{D}^{(t)^{-1}}$, we defined our PPR-based node embedding as following:

$$\boldsymbol{Z}^{(0)} = \sum_{l=0}^{\infty} \alpha(1-\alpha)^l \left(\boldsymbol{P}^{(0)}\right)^l \boldsymbol{X}^{(0)} = \alpha \left(\boldsymbol{I} - (1-\alpha)\boldsymbol{P}^{(0)}\right)^{-1} \boldsymbol{X}^{(0)}, \tag{2}$$

where $\boldsymbol{P}^{(0)}$ is the initial normalized adjacency matrix. To improve computational efficiency without sacrificing model effectiveness, we adopt the approximation $\boldsymbol{H}^{(0)}$ with the guarantee $||\boldsymbol{H}^{(0)} - \boldsymbol{Z}^{(0)}||_1 \leq n^{(0)}\epsilon$.

**Accuracy-guaranteed Embedding Update.** As mentioned, our objective is to efficiently propagate frequent updates into node embeddings from the preceding time step. The key rationale is that not all embeddings are significantly affected by updates, since an edge update usually impacts only nearby nodes (Mo & Luo, 2023). Motivated by this, single-snapshot methods such as Instant (Zheng et al., 2022) and IDOL (Zhu et al., 2024) update embeddings locally on the affected nodes. However, as shown in Lemma 1 and Proposition 1, our framework imposes a *strict theoretical requirement*: the approximated embedding $\boldsymbol{H}^{(t)}$ must satisfy $||\boldsymbol{H}^{(t)} - \boldsymbol{Z}^{(t)}||_1 \leq n^{(t)}\epsilon$ for each time step given the hyperparameter $\epsilon$. Unfortunately, existing methods cannot meet this condition, making them incompatible with our framework and leading to suboptimal prediction accuracy (see Section 4.4). To tackle this challenge, we draw on insights from recent works on Random Walk with Restart (RWR) (Yoon et al., 2018a) to compensate for embedding distance of each node across different time steps. Compared to existing approaches, our method provides a more robust adjustment mechanism, particularly in scenarios with significant changes in node degree, ensuring accuracy-guaranteed embedding updates. More details are provided in the Appendix A.8.

**From Embeddings to States.** After obtaining the parameter-free node embeddings $\boldsymbol{H}^{(t)}$, it is necessary to integrate the evolution information from these embedding sequences into the node states $\boldsymbol{M}^{(t)}$. However, existing mechanisms, such as RNNs and attention-based methods, often face challenges in balancing accuracy and efficiency, as highlighted in Tab. 1. Inspired by recent advancements in applying physical systems to sequence processing (Gu & Dao, 2023), we leverage the classical State Space Models (SSMs) (Kalman, 1960) to effectively summarize rich temporal information across different time steps. Specifically, given an edge update $e_{t+1}$ at time $t+1$, we update the state matrix $\boldsymbol{M}^{(t+1)}(u)$ for each $u \in \mathcal{V}^{(t+1)}$ with the following equations:

$$\boldsymbol{M}^{(t+1)} = \bar{\mathcal{A}} \cdot \boldsymbol{M}^{(t)} + \bar{\mathcal{B}} \cdot \boldsymbol{H}^{(t+1)}, \tag{3}$$

where $\bar{\mathcal{A}}, \bar{\mathcal{B}} \in \mathbb{R}^{F' \times F}$ are trainable parameters adhering to the standard formulations utilized in the literature (Gu & Dao, 2023; Gu et al.). The prediction of time $t+1$ is then generated through an MLP layer as $\boldsymbol{Y}^{(t+1)} = \text{MLP}(\boldsymbol{M}^{(t+1)})$. Compared with the methods (Li et al., 2024a;b) applying SSMs on temporal graph tasks, our approach is the first to prioritize efficiency of continuous prediction in practical dynamic scenarios. By ensuring the precision of the node embeddings, our method achieves a significant efficiency advantage over current algorithms while maintaining comparable prediction accuracy. In the following sections, we will explore the theoretical effectiveness of CODEN (see Sec. 3.2 and 3.3) and the underlying rationale behind its efficiency gap (see Appendix A.10).

## 3.2 EFFICIENT BATCH PROCESSING

In our previous discussions, we update the node states iteratively at each time step when new updates arrive, as outlined in Equ. 3. However, updates in TGNNs often occur in batches, especially in update-dense dynamic scenarios. In such cases, processing each update individually is associated with time complexity of $O(p)$ given $p$ updates, leading to prohibitive computation overhead. To address this challenge, we propose to sample intermediate embeddings across different snapshots.

**Lazy-sampling Strategy.** Since each update will only cause a minor change on the embedding space (Zheng et al., 2022; Zhu et al., 2024), we consider to skip the snapshots until there is a significant shift in the embedding space of $\boldsymbol{H}^{(t)}$. However, monitoring the shift of $\boldsymbol{H}^{(t)}$ requires sequential updates with each new arrival in the system, which significantly reduces efficiency. Therefore, we provide the following upper bound of the shift between $\boldsymbol{H}^{(t)}$ and $\boldsymbol{H}^{(t+1)}$ given an update at $t+1$ to estimate the potential shift and reduce the computational overhead:

**Lemma 1.** *If there exists an edge update at time $t+1$ and $||\boldsymbol{H}^{(t)} - \boldsymbol{Z}^{(t)}||_1 \leq n^{(t)}\epsilon$ holds for time $t$, the difference between $\boldsymbol{H}^{(t)}$ and $\boldsymbol{H}^{(t+1)}$ satisfies:*

$$||\boldsymbol{H}^{(t+1)} - \boldsymbol{H}^{(t)}||_1 \leq \frac{1-\alpha}{\alpha}\left\|\left(\boldsymbol{P}^{(t+1)} - \boldsymbol{P}^{(t)}\right) \cdot \boldsymbol{x}_{max}\right\|_1 + 2n^{(t)}\epsilon, \tag{4}$$

*where $\boldsymbol{x}_{max}$ is the row-wise maximum absolute value vector and the $i$-th entry of $\boldsymbol{x}_{max}$ is defined as: $\{\boldsymbol{x}_{max}\}_i = \max_{1 \leq j \leq F} |\boldsymbol{X}_{ij}^{(t)}|$.*

Interestingly, the R.H.S. of Equ 4 in Lemma 1 can be calculated efficiently as it only involves multiplication between a sparse matrix $\left(\boldsymbol{P}^{(t+1)} - \boldsymbol{P}^{(t)}\right)$ and vector $\boldsymbol{x}_{max}$. We set $\epsilon$ to $O(\frac{1}{(n^{(t)})^2})$ to ensure a small uppper bound. Since $||\boldsymbol{H}^{(t+p)} - \boldsymbol{H}^{(t)}||_1 \leq \sum_{i=0}^{p-1} ||\boldsymbol{H}^{(t+i+1)} - \boldsymbol{H}^{(t+i)}||_1$, we can accumulate the upper bound as indicated in Equ. 4 to further estimate the shift between $\boldsymbol{H}^{(t)}$ and $\boldsymbol{H}^{(t+p)}$ given a batch containing $p$ updates. We present this lazy-sampling strategy with the pseudo-code as shown in the Appendix A.8. The main idea of this algorithm is to accumulate the error bound for each edge update and delay the sampling of embeddings until the accumulated error exceeds a certain control error $\lambda$.

**Information Compression by Lazy-sampling**. As CODEN omits certain evolving information during the sampling process, it is crucial to assess its influence on the model's performance. By leveraging the upper bound on embedding distance mentioned above, we effectively control the extent of information loss from the evolving dynamics. Interestingly, we observe that CODEN is equally to compress the evolving information between time $t_k$ and $t_{k+1}$ of the graph into its state matrix $\boldsymbol{M}^{(t_{k+1})}$, preserving the essential temporal dependencies within the graph.

To simplify the illustration, we assume that there are two sampled embeddings between the prediction time $t_k$ and $t_{k+1}$ and the corresponding sampling time stamps are denoted as $\tau_1$ and $\tau_2$ ($t_k \leq \tau_1 \leq \tau_2 \leq t_{k+1}$). Then the distance of two adjacent sampled embedding using the lazy-sampling process are bounded such that $||\boldsymbol{H}^{(\tau_1)} - \boldsymbol{H}^{(\tau_2)}|| \leq \lambda$. Note that there are still multiple edge updates between $\tau_1$ and $\tau_2$, where the intermediate embeddings formed by these updates are omitted to enhance efficiency. Specifically, assuming there exist $p_s$ edges between two consecutively sampled embeddings $\boldsymbol{H}^{(\tau_1)}$ and $\boldsymbol{H}^{(\tau_2)}$, we demonstrate that CODEN can effectively incorporate these $p_s$ omitted embeddings as inputs to the SSM, as detailed in the following proposition:

**Proposition 1.** *If there are $p_s$ edges between the sampling time $\tau_1$ and $\tau_2$ and $||\boldsymbol{H}^{(t)} - \boldsymbol{Z}^{(t)}||_1 \leq n^{(t)}\epsilon$ holds for each time step $t$, then the input $\boldsymbol{H}^{(\tau_2)}$ saved in $\boldsymbol{M}^{(\tau_2)}$ can be regarded as the approximation of the normalized summation of all exact PPR-based embeddings between $\tau_1$ and $\tau_2$ such that:*

$$\left\|\boldsymbol{H}^{(\tau_2)} - \sum_{i=1}^{p_s}\sum_{l=0}^{\infty} \alpha\left((1-\alpha)^l \boldsymbol{P}^{(\tau_1+i)}\right)^l \boldsymbol{X}^{(\tau_1+i)}\Lambda_i\right\|_1 \leq \lambda,$$

*where $(\tau_1 + i)$ is the time step of the $i$-th edge between $\tau_1$ and $\tau_2$ and $\Lambda_i \in \mathbb{R}^{F \times F}$ is the hidden non-negative diagonal matrices saved in $\bar{\mathcal{A}}$ which satisfy $\sum_{i=1}^{p_s} \Lambda_i = \boldsymbol{I}$.*

Based on this proposition, the state update process is equivalent to inputting the normalized summation of the exact PPR-based embeddings between $\tau_1$ and $\tau_2$ into the SSM module. Specifically, the hidden diagonal matrices $\Lambda_i$ $(1 \le i \le p_s)$ will be implicitly modified during the training of the parameter $\bar{\mathcal{A}}$, which will in turn influence the amount of information retained in future time windows (Li et al., 2024a). In this scenario, we effectively compress the continuously evolving process, selectively retaining information that enhances prediction capabilities for downstream tasks. In Sec. 3.3, we will further reinforce this key insight by demonstrating its equivalence with attention-based algorithms.

### 3.3 DISCUSSION

**Theoretical Analysis on Equivalence.** As pioneers in applying SSM to temporal graphs, we further demonstrate our algorithm's equivalence to the kernel attention mechanism (Katharopoulos et al., 2020), an enhanced variant derived from the classical attention mechanism (Vaswani et al., 2017). Building on the duality of SSM (Dao & Gu, 2024b), we have the following proposition:

**Proposition 2.** CODEN *equivalently applies the kernel attention mechanism (Choromanski et al., 2020) for information compression which selectively masks the impact of certain snapshots.*

Note that in areas of computer vision (Darcet et al., 2024) and natural language processing (Xiao et al., 2024), the kernel attention mechanism is developed to mitigate the phenomenon of 'overattention' (Dao & Gu, 2024a) by selectively maintaining and degrading specific tokens (Xiao et al., 2024). In our case, recalling the significance of $\bar{\mathcal{A}}^{(i)}$ in Proposition 1, we establish a connection between CODEN and the kernel attention mechanism, selectively neglecting the status of certain snapshots and thus effectively controlling how much information is preserved over time. In essence, matrices $\bar{\mathcal{A}}^{(i)}$ effectively mask redundant information from the evolving process, thereby providing valuable contexts to the node state and significantly boosting performance in downstream tasks.

**Less over-smoothing and lower complexity.** With the equivalence mentioned earlier, an important question arises: *Given the equivalence, why do we still need the SSM module for* CODEN *when processing temporal graphs, especially in contrast to the attention module?* In order to further highlight the merits of incorporating the SSM mechanism into CODEN, we further provide a matched ablation variant named as CODEN-A. This variant retains all the features of CODEN but substitutes the SSM module with the *attention mechanism*, providing a direct comparison of their efficiency and effectiveness. Following the standard structure building of (Wang et al., 2021b; Yu et al., 2020), we compute the temporal dependency of node embeddings over multiple time steps by following the attention expression:

$$\boldsymbol{M}^{(0:t)} = \text{softmax}\left(\left(\boldsymbol{W}_q \boldsymbol{H}^{(0:t)}\right) \cdot \left(\boldsymbol{W}_k \boldsymbol{H}^{(0:t)}\right)/\sqrt{F'}\right) \cdot \left(\boldsymbol{W}_v \boldsymbol{H}^{(0:t)}\right),$$

where $\boldsymbol{W}_q, \boldsymbol{W}_k, \boldsymbol{W}_v \in \mathbb{R}^{F' \times F}$ are trainable parameters to calculate the queries, keys, and values, facilitating the modeling of interactions among different states across time steps. Specifically, we have the following proposition with respect to CODEN and CODEN-A:

**Proposition 3.** *Compared with* CODEN-A, CODEN *not only reduces time complexity but also mitigates over-smoothing under the Dirichlet energy measure.*

The proof and more detailed discussion on the superiority of CODEN can be found in the Appendix A.10. The above in-depth analysis shows that while CODEN produces an equivalent scheme with the attention mechanism, the non-trivial integration of SSM for temporal graph processing enhances both the efficiency and the quality of representations achieved by CODEN. Moreover, we believe this direct association between CODEN and the attention mechanism could shed light on the rationale behind the need for the application of SSM in temporal graphs.

## 4 EXPERIMENTS

In this section, we conduct comprehensive experiments to evaluate the key performances of CODEN against strong TGNN baselines. All experimental results are obtained with 10 runs on a Linux machine with an Intel(R) Xeon(R) Gold 6238R CPU @ 2.20GHz with 160GB RAM and an NVIDIA RTX A5000 with 24GB memory. Due to the limitation of space, we leave additional experimental results and related analysis in the Appendix A.12.

Table 3: The average, the best and the worst accuracy (%, denoted as Ave. and Bes. respectively) across all time steps. "OOM" stands for out of memory on a GPU with 24GB memory. The best results are ranked first , second for TGNN methods.

| Method | DBLP | | Tmall | | Reddit | | Patent | | Papers100M | |
|---|---|---|---|---|---|---|---|---|---|---|
| | Ave. | Bes. | Ave. | Bes. | Ave. | Bes. | Ave. | Bes. | Ave. | Bes. |
| GCN | $72.32_{\pm0.12}$ | $74.24_{\pm0.22}$ | $60.11_{\pm0.15}$ | $62.36_{\pm0.46}$ | $90.92_{\pm0.45}$ | $93.30_{\pm0.84}$ | $81.26_{\pm0.54}$ | $83.95_{\pm0.60}$ | OOM | OOM |
| GraphSage | $72.53_{\pm0.64}$ | $74.55_{\pm0.61}$ | $60.25_{\pm0.40}$ | $64.62_{\pm0.71}$ | $90.65_{\pm0.26}$ | $92.95_{\pm0.37}$ | $82.95_{\pm0.75}$ | $83.21_{\pm0.44}$ | OOM | OOM |
| SCARA | $73.57_{\pm0.29}$ | $75.68_{\pm0.66}$ | $61.32_{\pm0.43}$ | $63.25_{\pm0.11}$ | $91.65_{\pm0.24}$ | $92.29_{\pm0.50}$ | $82.44_{\pm0.32}$ | $83.55_{\pm0.29}$ | $62.22_{\pm0.46}$ | $63.40_{\pm0.21}$ |
| Instant | $74.25_{\pm0.21}$ | $75.22_{\pm0.44}$ | $61.22_{\pm0.87}$ | $64.26_{\pm0.43}$ | $92.10_{\pm0.67}$ | $92.34_{\pm0.29}$ | $81.95_{\pm0.37}$ | $82.99_{\pm0.59}$ | $62.75_{\pm0.17}$ | $64.15_{\pm0.35}$ |
| TGCN | $74.17_{\pm0.83}$ | $75.65_{\pm0.75}$ | $60.00_{\pm0.93}$ | $61.92_{\pm0.71}$ | $92.13_{\pm0.83}$ | $93.73_{\pm0.44}$ | $82.51_{\pm0.68}$ | $83.39_{\pm0.60}$ | OOM | OOM |
| ASTGCN | $75.04_{\pm0.54}$ | $76.7_{\pm0.19}$ | $61.80_{\pm0.72}$ | $66.59_{\pm0.62}$ | $91.66_{\pm0.69}$ | $93.79_{\pm0.81}$ | $82.52_{\pm0.70}$ | $83.36_{\pm0.56}$ | OOM | OOM |
| EvolveGCN | $73.27_{\pm0.48}$ | $76.09_{\pm0.54}$ | $64.27_{\pm0.63}$ | $66.05_{\pm0.39}$ | $91.33_{\pm0.53}$ | $93.50_{\pm0.27}$ | $82.74_{\pm0.46}$ | $83.51_{\pm0.22}$ | OOM | OOM |
| MPNN | $74.58_{\pm0.17}$ | $77.02_{\pm0.19}$ | $61.81_{\pm0.36}$ | $65.04_{\pm0.60}$ | $91.09_{\pm0.30}$ | $93.08_{\pm0.64}$ | $83.25_{\pm0.52}$ | $84.11_{\pm0.49}$ | OOM | OOM |
| CAWN | $73.56_{\pm0.27}$ | $75.58_{\pm0.67}$ | $62.34_{\pm0.39}$ | $65.35_{\pm0.21}$ | $91.54_{\pm0.76}$ | $93.53_{\pm0.65}$ | $82.21_{\pm0.36}$ | $83.62_{\pm0.72}$ | OOM | OOM |
| DNNTSP | $74.25_{\pm0.32}$ | $75.87_{\pm0.76}$ | $64.29_{\pm0.88}$ | $65.9_{\pm0.83}$ | OOM | OOM | OOM | OOM | OOM | OOM |
| SpikeNet | $74.83_{\pm0.58}$ | $75.72_{\pm0.71}$ | $64.21_{\pm0.40}$ | $66.02_{\pm0.66}$ | $92.19_{\pm0.41}$ | $93.83_{\pm0.37}$ | $81.86_{\pm0.63}$ | $82.52_{\pm0.47}$ | OOM | OOM |
| Zebra | $75.04_{\pm0.40}$ | $75.97_{\pm0.33}$ | $63.14_{\pm0.27}$ | $65.12_{\pm0.89}$ | $91.77_{\pm0.42}$ | $92.94_{\pm0.12}$ | $82.86_{\pm0.66}$ | $83.98_{\pm0.27}$ | OOM | OOM |
| TGL+TGN | $73.12_{\pm0.34}$ | $76.91_{\pm0.60}$ | $62.56_{\pm0.77}$ | $65.02_{\pm0.89}$ | $92.11_{\pm0.55}$ | $93.53_{\pm0.33}$ | $81.20_{\pm0.31}$ | $83.48_{\pm0.66}$ | $62.88_{\pm0.26}$ | $64.33_{\pm0.53}$ |
| DyGFormer | $73.88_{\pm0.54}$ | $76.79_{\pm0.29}$ | $63.04_{\pm0.30}$ | $65.94_{\pm0.33}$ | $91.54_{\pm0.61}$ | $93.32_{\pm0.62}$ | $82.16_{\pm0.47}$ | $83.59_{\pm0.25}$ | OOM | OOM |
| CODEN | $76.35_{\pm0.17}$ | $77.23_{\pm0.27}$ | $65.14_{\pm0.19}$ | $66.7_{\pm0.23}$ | $92.61_{\pm0.19}$ | $94.17_{\pm0.30}$ | $83.74_{\pm0.39}$ | $84.53_{\pm0.14}$ | $64.89_{\pm0.16}$ | $66.45_{\pm0.33}$ |

## 4.1 Datasets & Baseline Methods

**Datasets.** We adopt five representative real-world dynamic datasets: *DBLP* (Li et al., 2023a), *Tmall* (Lu et al., 2019), and three large-scale graphs, *Reddit* (Hamilton et al., 2017), *Patent* (Hall et al., 2001) and *Papers100M* (Hu et al., 2020). The statistics of the datasets are shown in Tab. 2. Within these datasets, the training, validation, and test sets are randomly allocated in proportions of 70%, 10%, and 20% respectively. To simulate scenarios that necessitate continuous prediction, we adopt the experimental framework outlined in (Zheng et al., 2022) and (Zhu et al., 2024), where the graph is segmented into an initial graph and $|T|$ batches of **edge sequences**. Then, each batch of edges will be added at distinct time steps in a evolving state. Although $T$ can be set to a theoretical upper bound of edge number $m$, we deliberately use a smaller $T$ for a cleaner demonstration. All methods are evaluated through a pipeline encompassing both training and inference.

**Baseline Methods.** We compare the proposed method CODEN with several state-of-the-art TGNN methods. They include: (i) the single-snapshot method Instant (Zheng et al., 2022); (ii) the RNN-based methods TGCN (Zhao et al., 2019), EvolveGCN (Pareja et al., 2020), MPNN (Panagopoulos et al., 2021), TGL+TGN (Zhou et al., 2022)[3] and CAWN (Wang et al., 2021d); (iii) the attention-based methods DNNTSP (Yu et al., 2020), DyGFormer (Yu et al., 2023a), and AST-GCN (Guo et al., 2019). Moreover, we also include the results of the recent competitive models SpikeNet (Li et al., 2023a) and Zebra (Li et al., 2023b). To this end, we include several strong baselines focusing on the single snapshot, such as classical GCN (Kipf & Welling, 2017), GraphSage (Hamilton et al., 2017), and the scalable methods SCARA (Liao et al., 2022). Specifically, we set the threshold $\epsilon = 1e^{-7}$, $\lambda = 0.1$, $F' = 16$, and $\alpha = 0.2$ by default in CODEN.

Table 2: Statistics of the datasets. $F$, $C$, and $|T|$ stand for the dimension of attributes, the number of classes, and the number of time steps to be predicted.

| **Datasets** | Nodes $n$ | Edges $m$ | $F$ | $C$ | $|T|$ |
|---|---|---|---|---|---|
| *DBLP* | 28,085 | 236,894 | 128 | 10 | 26 |
| *Tmall* | 577,314 | 4,807,545 | 80 | 5 | 19 |
| *Reddit* | 227,853 | 114,615,892 | 602 | 40 | 16 |
| *Patent* | 2,738,012 | 13,960,811 | 128 | 6 | 16 |
| *Papers100M* | 111,059,956 | 1,615,685,872 | 128 | 172 | 21 |

## 4.2 Accuracy Comparison

To simulate the evolving scenarios where the edge updates happen periodically, we first remove all edges from the graph and then progressively re-add them evenly in chronological order according to the time steps. With $|T|$ partitions of **edge sequences** in our setting, each dataset can yield $|T|$ distinct results. Note that this setting is substantially different from a general experimental setting in TGNNs, which typically focuses only on the fully formed final graph and fails to reflect the evolving process of CTDG. Collectively, our objective is to evaluate the performance of all compared methods in a scenario requiring continuous prediction.

**Overall statistic.** To synthetically evaluate the performance of all compared methods, we first provide detailed statistics including the average, best, and worst accuracy across all prediction times, as shown in Tab. 3. We separate the methods according to fact whether they incorporate the temporal

---

[3]We run TGL with 4 GPUs in parallel and adopt the most efficient backbone TGN (Rossi et al., 2020).

information. Since most of methods run out of memory on the *Papers100M* dataset, we mainly focus on the results from the other four datasets in Tab. 3.

**Temporal information is crucial for accurate prediction.** Methods designed for single snapshots, such as GCN, GraphSage, SCARA, and Instant, fall short compared to other TGNN approaches. This disparity arises because these methods focus exclusively on individual snapshots, overlooking the evolving dynamics within the graph. This underscores the critical role of temporal information in achieving more accurate predictions. Therefore, our comparison will primarily focus on methods that process temporal information in the following sections.

**CODEN exhibits desirable performance across various time steps.** We compare the accuracy performance of CODEN with other strong competitors. As shown in Fig. 1, we present the Micro-F1 scores on *Reddit* and *Patent* across different prediction times to demonstrate the accuracy comparison, and the results of other two datasets can be found in the Appendix A.12.3. As an overview, one key observation is indicated

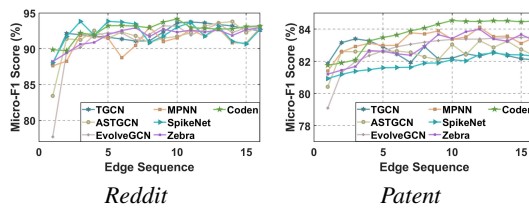

Figure 1: Micro-F1 scores comparison.

by these comparison results: CODEN *can match or even outperform the state-of-the-art methods in terms of prediction accuracy in most cases.* Furthermore, while other methods fluctuate across time steps, CODEN delivers more stable performance with steadily improving accuracy. We attribute this to two factors: (i) the embedding update algorithm maintains embedding quality throughout the evolving process, enhancing robustness during updates; and (ii) the lazy-sampling strategy combined with the SSM module enables effective compression that selectively retains valuable information while filtering noise.

### 4.3 EFFICIENCY COMPARISON

**CODEN achieves a speedup of up to $44.80\times$ in time consumption and finishes continuous predictions within 3.3 hours averagely on the billion-scale graph *Papers100M*.** To ensure a fair comparison of efficiency, we evaluate the total time consumption, encompassing *the network training (100 epochs), and inference*, which collectively constitute a single complete prediction. We report the overall training time and the inference time for each method, as shown in Tab. 4. The experimental results highlight the superiority of our model achieving efficiency through the learning phases. On the large-scale graph such as *Patent*, CODEN achieves $2.67 - 44.80\times$ acceleration in training time, along with improved inference speed.

Table 4: The total training time ($s$) and inference time ($s$, in parentheses).

| Method | DBLP | Tmall | Reddit | Patent | Papers100M |
|---|---|---|---|---|---|
| TGCN | 11.12 (0.04) | 214.23( 0.08) | $2.29K$ (12.16) | $4.62K$ (19.54) | OOM |
| ASTGCN | 439.56 (0.14) | $3.09K$ (1.50) | $51.03K$ (13.54) | $77.59K$ (34.58) | OOM |
| EvolveGCN | 8.43 (0.06) | $123.54$(0.08) | $2.61K$ (12.56) | $5.16K$ (19.45) | OOM |
| MPNN | 21.36 (0.04) | 303.65 (0.12) | $2.80K$ (13.10) | $6.43K$ (19.98) | OOM |
| CAWN | 9.25 (0.15) | $166.32$(0.19) | $3.20K$ (16.45) | $7.73K$ (24.00) | OOM |
| DNNTSP | $3.86K$ (0.10) | $17.35K$ (5.04) | OOM | OOM | OOM |
| SpikeNet | 255.11 (0.12) | 546.21 (0.55) | $3.91K$ (15.26) | $12.83K$ (29.56) | OOM |
| Zebra | 435.26 (0.14) | 946.28 (0.76) | $4.03K$ (15.28) | $20.34K$ (32.71) | OOM |
| TGL+TGN | 868.65 (0.15) | $1.96K$ (1.22) | $18.48K$ (40.99) | $38.56K$ (68.98) | $354K$ (150.45) |
| DyGFormer | $1.57K$ (0.19) | $9.68K$ (4.22) | $29.11K$ (28.15) | $58.51K$ (50.79) | OOM |
| CODEN | 6.00 (0.02) | 23.68(0.05) | $1.31K$ (8.15) | $1.73K$ (11.69) | $12.30K$ (20.15) |

To demonstrate this difference clearly, we record the detailed results on *Reddit* and *Patent* datasets across all time steps in Fig. 2, where the results of other two datasets are provided in the Appendix A.12.3. It is implied that all baselines experience an increase in training time as the number of edge updates grows, escalating rapidly with the data size. In contrast, CODEN exhibits only a slight increase in the time con-

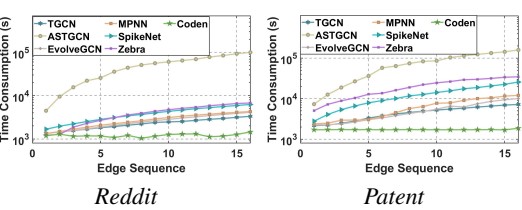

Figure 2: Training time comparison.

sumption (e.g., ranging from $1.71K$ s to $1.86K$ s on *Patent*), significantly improving training efficiency and reducing system workload. Furthermore, both CODEN and TGL+TGN are capable of performing continuous prediction on the *Papers100M* dataset, but CODEN significantly outperforms TGL+TGN

in terms of efficiency, remarkably handling the scale where other TGNN methods fail in memory limitations. We attribute CODEN's superior performance to two factors: (i) unlike baselines that recompute graph propagation from scratch, CODEN updates embeddings incrementally, avoiding redundant computation and reducing per-step training cost; and (ii) the SSM's linear recurrence preserves long-range historical information in node states. Together, these properties yield a more resource-friendly pipeline with faster computation.

## 4.4 ABLATION STUDY

We then perform an ablation study to demonstrate the contribution of each module in CODEN. We compare our model with its three variants: CODEN-S modifies CODEN by removing the temporal structure and employing an MLP classifier on the current node embeddings for straightforward prediction; CODEN-I inherits the structure of CODEN except for replacing the method of embedding

Table 5: The average accuracy (%) and time consumption (s) across all time steps for each variant of CODEN.

| Method | DBLP | | Tmall | | Reddit | | Patent | |
|--------|------|------|-------|-------|--------|--------|--------|--------|
| | Ave. | Tim. | Ave. | Tim. | Ave. | Tim. | Ave. | Tim. |
| CODEN-S | 74.88 | 0.04 | 61.24 | 0.18 | 91.11 | 6.86 | 81.78 | 11.99 |
| CODEN-I | 74.21 | 0.06 | 61.07 | 0.22 | 91.04 | 10.55 | 81.65 | 13.69 |
| CODEN-A | 76.07 | 3.19 | 64.69 | 27.69 | 91.67 | 412.38 | 83.09 | 672.64 |
| CODEN | 76.35 | 0.08 | 65.14 | 0.28 | 92.38 | 12.10 | 83.74 | 17.26 |

update with the invariant-based scheme in existing related works (Zheng et al., 2022; Guo et al., 2022; Zhu et al., 2024); CODEN-A is the ablation variant utilizing a attention structure for message passing in Sec. 3.3. As shown in Tab. 5, CODEN-S is the most efficient variant since it removes dependencies across time, but the performance gap to CODEN confirms the importance of temporal information. CODEN-A leverages historical context through attention and yields more accurate representations than CODEN-S, while its high complexity negates efficiency gains and its accuracy still lags behind CODEN, likely due to over-smoothing (Lemma 6). To this end, the invariant-based scheme is incompatible with our framework and can even underperform CODEN-S. Without accuracy guarantees, it produces embedding-scale discrepancies across time steps, and aggregating such inconsistent information disturbs node states, degrading prediction accuracy.

### 4.4.1 EFFECT OF HYPERPARAMETERS

We also examine the sensitivity of CODEN to hyperparameters. First, we study the sampling threshold $\lambda$, which controls the embedding sequences in the SSM module and affects training time. Second, similar to the gating mechanism in RNNs, the hidden dimension $F'$ influences the degree of information compression. We vary these settings and report results on the *DBLP* dataset.

**Threshold $\lambda$.** As shown in Fig. 3, we report the average training time and accuracy across time steps by varying the sampling threshold $\lambda$. A clear trade-off emerges: larger $\lambda$ values reduce training time but degrade accuracy. Once $\lambda$ exceeds a certain level (e.g., 0.05), the model samples only a few embeddings and degrade the performance, underscoring the importance of dynamic modeling for accurate prediction.

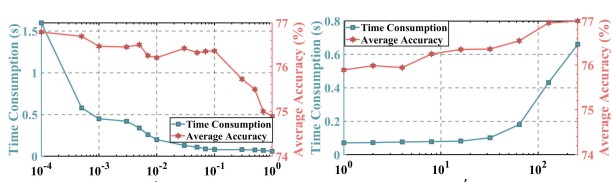

Figure 3: The average training time and the average accuracy when setting different $\lambda$ and $F'$ on *DBLP* dataset.

**Hidden dimension $F'$.** The hidden dimension $F'$ in SSM controls the shape of selected input information. As shown in Fig. 3, varying $F'$ has little effect on prediction accuracy, with increases beyond 16 yielding only marginal gains (less than $0.5\%$). However, overly large values ($F' > 64$) degrade training efficiency. We therefore set $F' = 16$ as the default for optimal performance.

## 5 CONCLUSION

In this work, we present CODEN, an innovative framework intending to improve the performance of TGNNs in a dynamic scenario requiring continuous predictions. CODEN performs a unique state-updating mechanism, where the node embeddings are updated incrementally and facilitate the compression of historical information in the state. Theoretically, CODEN can provide an accuracy guarantee after the embedding update and approximately achieve information compression of the continuously evolving process. Our extensive experimental results demonstrate CODEN's superior performance in both efficacy and efficiency when compared with state-of-the-art methods.

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

# A  APPENDIX

## A.1  NOTATIONS

Table 6: Frequently used notations in this paper.

| Notations | Descriptions |
|---|---|
| $\mathcal{G}^{(t)}$ | Directed graph at time $t$ |
| $n^{(t)}, m^{(t)}$ | Numbers of nodes and edges at time $t$ |
| $F$ | Dimension of the node attribute |
| $\boldsymbol{X}^{(t)}$ | The node attribute matrix and $\boldsymbol{X}^{(t)} \in \mathbb{R}^{n^{(t)} \times F}$ |
| $\boldsymbol{x}_i^{(t)}$ | The attribute vector of the $i$-th dimension |
| $\mathcal{N}_{out}^{(t)}(v), \mathcal{N}_{in}^{(t)}(v)$ | The out-neighbors and in-neighbors of $v$ |
| $e_t = \{u_t, v_t\}, \Gamma$ | The update event and the event set |
| $\boldsymbol{r}^{(t)}$ | Residue vector at time $t$ and $\boldsymbol{r}^{(t)} \in \mathbb{R}^{n^{(t)} \times 1}$ |
| $\boldsymbol{h}^{(t)}$ | Reserve vector at time $t$ and $\boldsymbol{h}^{(t)} \in \mathbb{R}^{n^{(t)} \times 1}$ |
| $\boldsymbol{H}^{(t)}$ | The node embedding matrix at time $t$ |
| $\boldsymbol{M}^{(t)}$ | The state matrix at time $t$ |
| $\bar{\mathcal{A}}, \bar{\mathcal{B}}, \mathcal{C}$ | The network parameters in SSM |
| $\alpha$ | Teleport probability of random walks |
| $\lambda$ | Threshold in to control the embedding distance |

## A.2  PROOF OF LEMMA 1

*Proof.* According to the definition of 1-Norm of the matrix, we have:

$$||\boldsymbol{H}^{(t+1)} - \boldsymbol{H}^{(t)}||_1 = \max_{f \in F} \sum_u |\boldsymbol{h}^{(t+1)}(u)) - \boldsymbol{h}^{(t)}(u))|$$

$$\leq \max_{f \in F} \sum_u |\boldsymbol{z}^{(t+1)}(u) - \boldsymbol{z}^{(t)}(u)) + 2\epsilon|$$

We can express $\boldsymbol{Z}^{(t)}$ as the result of global smoothness (Ma et al., 2021) which satisfies:

$$\boldsymbol{Z}^{(t)} = \left(\boldsymbol{I} + c\boldsymbol{L}^{(t)}\right)^{-1} \boldsymbol{X}^{(t)}, \tag{5}$$

where $c = \frac{1}{\alpha} - 1$ and $\boldsymbol{L}^{(t)} = \boldsymbol{I} - \boldsymbol{P}^{(t)}$ is the Laplacian matrix and denotes the topological status of graph $\mathcal{G}^{(t)}$. Hence we have:

$$||\boldsymbol{H}^{(t+1)} - \boldsymbol{H}^{(t)}||_1$$

$$\leq \max_{f \in F} \left\|\left(\left(\boldsymbol{I} + c\boldsymbol{L}^{(t+1)}\right)^{-1} - \left(\boldsymbol{I} + c\boldsymbol{L}^{(t)}\right)^{-1}\right) \boldsymbol{x}^{(t)}\right\|_1 + 2n\epsilon$$

$$\leq c \max_{f \in F} \left\|\left(\boldsymbol{I} + c\boldsymbol{L}^{(t+1)}\right)^{-1} \left(\boldsymbol{L}^{(t+1)} - \boldsymbol{L}^{(t)}\right) \left(\boldsymbol{I} + c\boldsymbol{L}^{(t)}\right)^{-1} \boldsymbol{x}^{(t)}\right\|_1$$

$$+ 2n\epsilon \leq c \max_{f \in F} \left\|\left(\boldsymbol{P}^{(t+1)} - \boldsymbol{P}^{(t)}\right) \boldsymbol{x}^{(t)}\right\|_1 + 2n\epsilon$$

$$\leq \frac{1 - \alpha}{\alpha} \|\Delta\boldsymbol{P} \cdot \boldsymbol{x}_{max}\|_1 + 2n\epsilon,$$

where $\boldsymbol{x}_{max}$ is the row-wise maximum absolute value vector and $i$-th entry of $\boldsymbol{x}_{max}$ is defined as: $\{\boldsymbol{x}_{max}\}_i = \max_{1 \leq j \leq n} |\boldsymbol{X}_{ij}|$. Proof finished.

$\square$

## A.3  PROOF OF PROPOSITION 1

*Proof.* We assume there come $p$ edge updates between $t$ and $t + 1$. Since we can not access the concrete timestamps of these $p$ edges, we can assume $k$-th each edge arrives at the system at a hidden timestamp $t + \tau_k$ ($1 \leq k \leq p$) without loss of generality. Specifically, we have $t + \tau_p = t + 1$.

According to (Li et al., 2024a), the complete evolving process from $t$ to $t+1$ can be formulated as:

$$\boldsymbol{M}^{(t+1)} = \bar{\mathcal{A}}\boldsymbol{M}^{(t)} + \bar{\mathcal{B}}\sum_{k=1}^{p}\sum_{l=0}^{\infty}\alpha\left((1-\alpha)^l\boldsymbol{P}^{(t+\tau_k)}\right)^l\boldsymbol{X}\Lambda_k$$

$$= \bar{\mathcal{A}}\boldsymbol{M}^{(t)} + \bar{\mathcal{B}}\sum_{k=1}^{p}\alpha\left(\boldsymbol{I} - (1-\alpha)\boldsymbol{P}^{(t+\tau_k)}\right)^{-1}\boldsymbol{X}\Lambda_k.$$

When setting $\alpha = \frac{1}{c+1}$, we have:

$$\boldsymbol{M}^{(t+1)} = \bar{\mathcal{A}}\boldsymbol{M}^{(t)} + \bar{\mathcal{B}}\sum_{k=1}^{p}\left(\boldsymbol{I} + c\boldsymbol{L}^{(t+\tau_k)}\right)^{-1}\boldsymbol{X}\Lambda_k,$$

where $\boldsymbol{L}^{(t)}$ is the Laplacian matrix at time $t$. Given the exact PPR embedding $\left(\boldsymbol{I} + c\boldsymbol{L}^{(t+1)}\right)^{-1}\boldsymbol{X}$ at time $t+1$, we have:

$$\left\|\left(\boldsymbol{I} + c\boldsymbol{L}^{(t+1)}\right)^{-1}\boldsymbol{X} - \sum_{k=1}^{p}\left(\boldsymbol{I} + c\boldsymbol{L}^{(t+\tau_k)}\right)^{-1}\boldsymbol{X}\Lambda_k\right\|$$

$$= \left\|\left(\boldsymbol{I} + c\boldsymbol{L}^{(t+1)}\right)^{-1}\boldsymbol{X}\sum_{k=1}^{p}\Lambda_k - \sum_{k=1}^{p}\left(\boldsymbol{I} + c\boldsymbol{L}^{(t+\tau_k)}\right)^{-1}\boldsymbol{X}\Lambda_k\right\|$$

$$= \left\|\sum_{k=1}^{p}\left(\left(\boldsymbol{I} + c\boldsymbol{L}^{(t+1)}\right)^{-1} - \left(\boldsymbol{I} + c\boldsymbol{L}^{(t+\tau_k)}\right)^{-1}\right)\boldsymbol{X}\Lambda_k\right\|$$

$$\leq \left\|\sum_{k=1}^{p}\left(\boldsymbol{I} + c\boldsymbol{L}^{(t+1)}\right)^{-1}\left(c\boldsymbol{L}^{(t+1)} - c\boldsymbol{L}^{(t+\tau_k)}\right)\left(\boldsymbol{I} + c\boldsymbol{L}^{(t+\tau_k)}\right)^{-1}\boldsymbol{X}\Lambda_k\right\|$$

$$\leq c\left\|\sum_{k=1}^{p}\left(\boldsymbol{I} + c\boldsymbol{L}^{(t+1)}\right)^{-1}\left(\boldsymbol{L}^{(t+1)} - \boldsymbol{L}^{(t+\tau_k)}\right)\left(\boldsymbol{I} + c\boldsymbol{L}^{(t+\tau_k)}\right)^{-1}\boldsymbol{X}\right\|\left\|\sum_{k=1}^{p}\Lambda_k\right\|$$

$$\leq c\sum_{k=1}^{p}\frac{\left\|\left(\boldsymbol{L}^{(t+1)} - \boldsymbol{L}^{(t+\tau_k)}\right)\boldsymbol{X}\right\|_2}{\left(1 + c\lambda_1(\boldsymbol{L}^{(t+1)})\right)\left(1 + c\lambda_1(\boldsymbol{L}^{(t+\tau_k)})\right)}$$

$$\leq c\sum_{k=1}^{p}\left\|\left(\boldsymbol{L}^{(t+1)} - \boldsymbol{L}^{(t+\tau_k)}\right)\boldsymbol{X}\right\| \leq \lambda$$

$$\square$$

## A.4  PROOF OF PROPOSITION 2

*Proof.* Given $\boldsymbol{M}^{(0)} = \bar{\mathcal{A}}^{(0)}\cdot\boldsymbol{0} + \bar{\mathcal{B}}^{(0)}\cdot\boldsymbol{H}^{(0)} = \bar{\mathcal{B}}^{(0)}\boldsymbol{H}^{(0)}$ where $\bar{\mathcal{A}}^{(t)}$ and $\bar{\mathcal{B}}^{(t)}$ denote the parameter metrices at time $t$. Following the state update principle [4] of Equ. 3, the state matrix at $t$ can be formulated as:

$$\boldsymbol{M}^{(t)} = \bar{\mathcal{A}}^{(t)}\bar{\mathcal{A}}^{(t-1)}...\bar{\mathcal{A}}^{(1)}\bar{\mathcal{B}}^{(0)}\boldsymbol{H}^{(0)} + \bar{\mathcal{A}}^{(t)}\bar{\mathcal{A}}^{(t-1)}...\bar{\mathcal{A}}^{(2)}\bar{\mathcal{B}}^{(1)}\boldsymbol{H}^{(1)} +$$

$$... + \bar{\mathcal{A}}^{(t)}\bar{\mathcal{A}}^{(t-1)}\bar{\mathcal{B}}^{(t-2)}\boldsymbol{H}^{(t-2)} + \bar{\mathcal{A}}^{(t)}\bar{\mathcal{B}}^{(t-1)}\boldsymbol{H}^{(t-1)} + \bar{\mathcal{B}}^{(t)}\boldsymbol{H}^{(t)}$$

$$= \sum_{s=0}^{t}\prod_{i=s+1}^{t}\bar{\mathcal{A}}^{(i)}\bar{\mathcal{B}}^{(s)}\boldsymbol{H}^{(s)}, \tag{6}$$

where we define $\prod_{i=t+1}^{t}\bar{\mathcal{A}}^{(i)} = \boldsymbol{I}$. When we vectorize the prediction result over time $[0,t]$ and restructure the mathematical expression, we can derive the formula representing our prediction across all time steps:

$$\boldsymbol{Y}^{(0:t)} = \left(\boldsymbol{L} \circ \boldsymbol{Q}\boldsymbol{K}^{\top}\right)\cdot\boldsymbol{V}, \tag{7}$$

where (i) $\boldsymbol{L}$ is a lower-triangular matrix and $\boldsymbol{L}_{uv} = \prod_{i=v+1}^{u}\bar{\mathcal{A}}^{(i)}$; (ii) $\boldsymbol{Q}$ is a lower-triangular matrix and $\boldsymbol{Q}_{uv} = \mathcal{C}^{(u)}$, where $\mathcal{C}$ is the parameters of MLP$(\cdot)$; (iii) $\boldsymbol{K}$ is a diagonal matrix and $\boldsymbol{K}_{uu} = \bar{\mathcal{B}}^{(u)}$; (iv) $\boldsymbol{V}$ is the vectorized node embedding generated at all time steps and $\boldsymbol{V}(u) = \boldsymbol{H}^{(u)}$.

---

[4] Without the loss of generalization, here we simplify the notation and assume the sample time corresponds with the prediction time for a clear presentation.

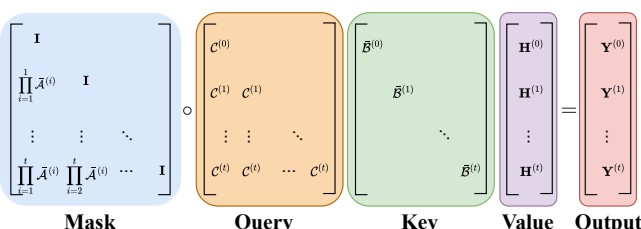

Figure 4: The equivalent kernel attention in CODEN.

As illustrated in Fig. 4, we observe that the fundamental methodology of CODEN for processing temporal information is fully aligned with the definition of the kernel attention mechanism (Choromanski et al., 2020), which is developed to mitigate the "over attention" phenomenon (Dao & Gu, 2024a) in computer vision (Darcet et al., 2024) and natural language processing (NLP) areas (Xiao et al., 2024). Upon revisiting the motivation behind kernel attention in NLP, where it is crucial to selectively maintain and degrade specific tokens (Xiao et al., 2024), it becomes apparent that the trainable matrices $\bar{\mathcal{A}}^{(i)}$ ($0 \leq i \leq t$) play a pivotal role. □

## A.5 PROOF OF LEMMA 6

*Proof.* Based on the definition of the Dirichlet Energy and simplifying $\boldsymbol{I} - \boldsymbol{A}^{(t)\top}\boldsymbol{D}^{(t)-1}$ as $\Delta$, we have:

$$\mathbb{DE}(\boldsymbol{M}_A^{(t)}) = \mathbb{DE}\left(\text{softmax}\left(\frac{\left(\boldsymbol{W}_q\boldsymbol{H}^{(t)}\right)\cdot\left(\boldsymbol{W}_k\boldsymbol{H}^{(s)}\right)^\top}{\sqrt{F'}}\right)_{0:t}\cdot\left(\boldsymbol{W}_v\boldsymbol{H}^{(s)}\right)_{0:t}\right).$$

To simplify the demonstration, we denote

$$\boldsymbol{S}_t = \text{softmax}\left(\left(\boldsymbol{W}_q\boldsymbol{H}^{(t)}\right)\cdot\left(\boldsymbol{W}_k\boldsymbol{H}^{(s)}\right)^\top\right)_{0:t}$$

. Since $\boldsymbol{S}_t$ is a Row-Stochastic matrix where the sum of each row equals to 1, we can obtain the maximum eigenvalue $\sigma_{max}(\boldsymbol{S}_t)$ of $\boldsymbol{S}_t$ as $\sigma_{max}(\boldsymbol{S}_t) \leq 1$. Then we have:

$$\mathbb{DE}(\boldsymbol{M}_A^{(t)}) = tr\left(\boldsymbol{S}_t\left(\boldsymbol{H}^{(s)}\right)_{0:t}\Delta\boldsymbol{S}_t^\top\left(\boldsymbol{H}^{(s)}\right)_{0:t}^\top\right)/F$$

$$= tr\left(\left(\boldsymbol{H}^{(s)}\right)_{0:t}\Delta\left(\boldsymbol{H}^{(s)}\right)_{0:t}^\top\boldsymbol{S}_t\boldsymbol{S}_t^\top\right)/F$$

$$\leq tr\left(\left(\boldsymbol{H}^{(s)}\right)_{0:t}\Delta\left(\boldsymbol{H}^{(s)}\right)_{0:t}^\top\right)\sigma_{max}(\boldsymbol{S}_t\boldsymbol{S}_t^\top)/F$$

$$\leq tr\left(\left(\boldsymbol{H}^{(s)}\right)_{0:t}\Delta\left(\boldsymbol{H}^{(s)}\right)_{0:t}^\top\right)/F$$

$$= \sum_{s=0}^t tr\left(\left(\boldsymbol{H}^{(s)}\right)\left(\boldsymbol{H}^{(s)}\right)^\top\Delta\right)/F$$

Following the similar derivation, we can obtain the Dirichlet Energy $\mathbb{DE}(\boldsymbol{M}_C^{(t)})$ as:

$$\mathbb{DE}(\boldsymbol{M}_C^{(t)}) \geq \sum_{s=0}^t tr\left(\prod_{i=s+1}^t \bar{\mathcal{A}}^{(i)}\bar{\mathcal{B}}^{(s)}\left(\boldsymbol{H}^{(s)}\right)\left(\boldsymbol{H}^{(s)}\right)^\top\Delta\right)$$

$$\geq \sum_{s=0}^t \sigma_{min}(\prod_{i=s+1}^t \bar{\mathcal{A}}^{(i)}\bar{\mathcal{B}}^{(s)})tr\left(\left(\boldsymbol{H}^{(s)}\right)\left(\boldsymbol{H}^{(s)}\right)^\top\Delta\right)$$

$$\geq \mathbb{DE}(\boldsymbol{M}_A^{(t)}),$$

which holds when $F\cdot\sigma_{min}^2(\prod_{i=s+1}^t \bar{\mathcal{A}}^{(i)}\bar{\mathcal{B}}^{(s)}) \geq 1$ for $0 \leq s \leq t$.

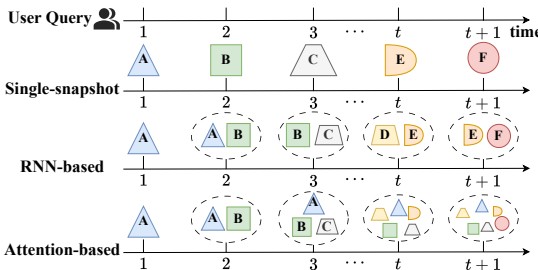

Figure 5: The comparison between different TGNN paradigms, where each color denotes a different status of the graph.

In the following, we provide an preliminary lemma to indicate how we make $F \cdot \sigma_{min}^2(\prod_{i=s+1}^{t} \bar{\mathcal{A}}^{(i)}\bar{\mathcal{B}}^{(s)}) \geq 1$ hold for $0 \leq s \leq t$.

**Lemma 2.** *For each time step $1 \leq i \leq T$, let $\bar{\mathcal{A}}^{(i)}$ be a learnable square matrix, and for each $0 \leq s \leq T-1$, let $\bar{\mathcal{B}}^{(s)}$ be another learnable square matrix. Define $P_{s,t} = \prod_{i=s+1}^{t} \bar{\mathcal{A}}^{(i)}\bar{\mathcal{B}}^{(s)}, 0 \leq s \leq t \leq T$. Assume every factor satisfies $\sigma_{\min}(W) \geq \gamma > 0$. If $\gamma \geq F^{-1/(2T)}$, then for all $0 \leq s \leq t \leq T$, $F \cdot \sigma_{\min}^2(P_{s,t}) \geq 1$.*

*Proof.* For any two real matrices $A$ and $B$ of compatible shapes,
$$\sigma_{\min}(AB) \geq \sigma_{\min}(A)\sigma_{\min}(B), \tag{8}$$
because $\|ABx\|_2 = \|A(Bx)\|_2 \geq \sigma_{\min}(A)\|Bx\|_2 \geq \sigma_{\min}(A)\sigma_{\min}(B)\|x\|_2$ for every unit vector $x$.

Apply (8) recursively to $P_{s,t} = \prod_{i=s+1}^{t} \bar{\mathcal{A}}^{(i)}\bar{\mathcal{B}}^{(s)}, 0 \leq s \leq t \leq T$, to obtain
$$\sigma_{\min}(P_{s,t}) \geq \prod_{i=s+1}^{t} \sigma_{\min}(\bar{\mathcal{A}}^{(i)})\sigma_{\min}(\bar{\mathcal{B}}^{(s)}). \tag{9}$$

Because every matrix $W \in \mathcal{W} = \{\bar{\mathcal{A}}^{(i)}, \bar{\mathcal{B}}^{(s)} \mid 1 \leq i \leq T, 0 \leq s \leq T-1\}$ satisfies that $\sigma_{\min}(W) \geq \gamma$, $\sigma_{\min}(P_{s,t}) \geq \gamma^{t-s+1}$. Since $t-s+1 \leq T$, we need to guarantee that $\sigma_{\min}(P_{s,t}) \geq \gamma^T$. Therefore $F \cdot \sigma_{\min}^2(P_{s,t}) \geq F \cdot \gamma^{2T} \geq 1$ when $\gamma \geq F^{-1/(2T)}$. □

Because we set $\gamma = c\, F^{-1/(2T)}(c > 1)$ and $\sigma_{\min}(W)$ will be close to $\gamma$, we can have $F \cdot \sigma_{\min}^2(P_{s,t}) > 1$ in our practical experiments.

□

## A.6 RELATED WORK

### A.6.1 SINGLE-SNAPSHOT METHODS

The single-snapshot GNN methods directly utilize the node embedding as the node states, which will be rapidly updated based on the new interactions. Specifically, single-snapshot GNN methods aim to transmit the updates into node embeddings $\boldsymbol{H}^{(t+1)}$ from $\boldsymbol{H}^{(t)}$ with the minimal time cost following:
$$\boldsymbol{H}^{(t+1)} = \text{MSG}\left(\mathcal{G}^{(t)}, \boldsymbol{H}^{(t)}, e_{t+1}\right). \tag{10}$$
To facilitate the incremental updating based on these pre-computed node embeddings, Instant (Zheng et al., 2022), DynAnom (Guo et al., 2022) and IDOL (Zhu et al., 2024) explore the invariant rules of graph propagation and conduct an invariant-based algorithm of Personalized PageRank (PPR) to refresh the node embedding locally. The dominant update complexity of these methods can be bounded as $O(p \sum_{i=1}^{F} \frac{\|\boldsymbol{x}_i^{(t)}\|_1}{\epsilon n^{(t)}})$ given $p$ updates and $\epsilon$ approximation error (Zheng et al., 2022). However, only focusing on the current snapshot will miss significant interactions of past time steps, leading to sub-optimal prediction results.

### A.6.2 RNN-BASED METHODS

RNN-based methods are generally based on the classical RNN (Cho et al., 2014), GRU (Chung et al., 2014), LSTM (Hochreiter & Schmidhuber, 1997) algorithm, etc., which simply use both the current input and the previous hidden state to iteratively capture temporal dependencies. For example, the GRU algorithm updates the node states at time $t + 1$ as:

$$\boldsymbol{Z}^{(t+1)} = \text{sigmoid}\left(\boldsymbol{W}_Z \boldsymbol{H}^{(t+1)} + \boldsymbol{U}_Z \boldsymbol{M}^{(t)} + \boldsymbol{B}_Z\right)$$

$$\boldsymbol{R}_t = \text{sigmoid}\left(\boldsymbol{W}_R \boldsymbol{H}^{(t+1)} + \boldsymbol{U}_R \boldsymbol{M}^{(t)} + \boldsymbol{B}_R\right)$$

$$\widetilde{\boldsymbol{H}}^{(t+1)} = \tanh\left(\boldsymbol{W}_H \boldsymbol{H}^{(t+1)} + \boldsymbol{U}_H \left(\boldsymbol{R}_t \circ \boldsymbol{M}^{(t)}\right) + \boldsymbol{B}_H\right)$$

$$\boldsymbol{M}^{(t+1)} = \left(1 - \boldsymbol{Z}^{(t+1)}\right) \circ \boldsymbol{M}^{(t)} + \boldsymbol{Z}^{(t+1)} \circ \widetilde{\boldsymbol{H}}^{(t+1)},$$

where $\boldsymbol{W}, \boldsymbol{U}, \boldsymbol{B}$ denote the trainable parameters of the linear layer and $\boldsymbol{H}^{(t+1)}$ can be updated using the MSG$(\cdot)$. Specifically, TGCN (Zhao et al., 2019) and EvolveGCN (Pareja et al., 2020) and incorporate the Graph Convolutional Network (GCN)(Kipf & Welling, 2017) as the MSG$(\cdot)$ function to regenerate the node embeddings while coupling with an RNN-based module to learn temporal node representations. Similarly, MPNN (Panagopoulos et al., 2021) transforms the node embeddings at different time steps into an RNN module and then captures the long-range dependency in the final hidden state. This category of method generally requires $O(Km^{(t)}F)$ and $O(pn^{(t)}F^2)$ to update the node embeddings and states [5], respectively. Due to their iterative structure, RNN-based methods can efficiently update node states. However, the simplistic recurrence mechanism often leads to difficulties in retaining historical information, especially as the graph size and temporal scope expand (Li et al., 2024a; Gers et al., 2000).

### A.6.3 ATTENTION-BASED METHODS

To address the forgetting problem of RNNs, attention-based methods rely on the attention mechanism and abstain from using recurrence form, which encodes the position of sequences and enables the efficient information flow from past to current representations. We take the representative work APAN (Wang et al., 2021c) as an example to demonstrate the core mechanism of this category. Considering matrices $\boldsymbol{Q} \in \mathbb{R}^{n \times F}$ denoted as "query", $\boldsymbol{K} \in \mathbb{R}^{n \times F}$ denoted as "keys", and $\boldsymbol{V}^{n \times F}$ denoted as "values", the classical attention algorithms perform the following computation to obtain the optimized embeddings:

$$\boldsymbol{M}^{(t+1)} = \text{Attn}(\boldsymbol{Q}, \boldsymbol{K}, \boldsymbol{V}) = \text{softmax}\left(\frac{\boldsymbol{Q}\boldsymbol{K}^\top}{\sqrt{F}}\right)\boldsymbol{V},$$

$$\boldsymbol{Q} = \boldsymbol{H}^{(t+1)}\boldsymbol{W}_q, \boldsymbol{K} = \boldsymbol{M}^{(t)}\boldsymbol{W}_k, \boldsymbol{V} = \boldsymbol{M}^{(t)}\boldsymbol{W}_v,$$

where $\boldsymbol{W}_q, \boldsymbol{W}_k, \boldsymbol{W} \in \mathbb{R}^{F \times F'}$ are the network parameters. The dot-product term $\left(\frac{\boldsymbol{Q}\boldsymbol{K}^\top}{\sqrt{F}}\right)$ takes the role of weighting the interactions between entity "query-key" pairs. A higher value within this term increases the contribution of $\boldsymbol{V}$ to the embedding space. Thus, attention-based methods create the expressive attention score to capture the relationship between the current embedding and the state of the last time step. Following this intuition, DySat (Sankar et al., 2020) employs the generalized GAT module (Veličković et al., 2018) to integrate the embeddings of a single node from different time steps to generate its refreshed one. ASTGCN (Guo et al., 2019) and DNNTSP (Yu et al., 2020) further incorporate the attention mechanism to capture the spatial and temporal dependency for enhanced representation quality. For each time step, these methods generally require $O(T\left(n^{(t)}\right)^2 F)$ time complexity to calculate the final output given $T$ time step. While attention mechanisms can retain the most relevant parts of the sequences to avoid the forgetting issue, they can become computationally expensive with frequent updates (Thomas et al., 2020).

### A.6.4 OTHER TGNN METHODS.

(i) *SNN-based methods.* Another typical mechanism of this category is based on the biological Spiking Neural Networks (SNNs), which simulate the brain behaviors and maintain the membrane potential given the data sequences. SpikeNet (Li et al., 2023a) retrieves the node embeddings from

---

[5]For a clear presentation, we assume the dimension of node state $F' = F$

multiple time steps and finally generates the prediction results by the spike firing process. Dy-SIGN (Yin et al., 2024) incorporates SNNs mechanism into dynamic graphs to mitigate the information loss and memory consumption problem. Nevertheless, the demand for multiple simulation steps to generate reliable embeddings can significantly degrade the efficiency of these TGNN methods. (ii) *SSM-based methods.* There are also some works which employs the SSM mechanism on temporal graphs. For example, STG-Mamba (Li et al., 2024b) formulates the feature stream of each node as the long-term context, which improves the embedding quality for feature-varying graphs. Graph-SSM (Li et al., 2024a) addresses the unobserved graph mutations between consecutive snapshots, and achieves an effective discretization with long-term information. However, these methods only focus on the discrete-time dynamic graph and fail to model the continuous topology changes as the graph evolves. Furthermore, directly adapting these methods will incur significant computational overhead as the graph evolves, creating a gap between current algorithms and continuous prediction in practical dynamic scenarios.

### A.6.5 STATE SPACE MODELS

In recent years, structured State Space Models (S4) (Gu et al.; 2021) have emerged as promising frameworks for modeling long-distance sequences, offering the advantage of only a linear increase in computational cost. Given the input series $x(t) \in \mathbb{R}$, structured State Space Models (S4) (Gu et al.; 2021) formulates the state variable series $h(t) \in \mathbb{R}$ and the output series $y(t) \in \mathbb{R}$ using the following equations:

$$\begin{cases} h'(t) & = \mathcal{A}h(t) + \mathcal{B}x(t) \\ y(t) & = \mathcal{C}h(t), \end{cases} \tag{11}$$

$$\begin{cases} h_t & = \bar{\mathcal{A}}h_{t-1} + \bar{\mathcal{B}}x_t \\ y_t & = \mathcal{C}h_t, \end{cases} \tag{12}$$

where $(\mathcal{A}, \mathcal{B}, \mathcal{C}, \bar{\mathcal{A}}, \bar{\mathcal{B}})$ are trainable parameters. Here Equ. 11 and 12 are the continuous and discretization process, respectively. Compared with the RNN-based or attention-based mechanism, S4's strength lies in its adherence to a linear mechanism, which guarantees enhanced stability control (Li et al., 2024a). This facilitates effective long-term modeling of sequences through meticulous initialization of state space layer parameters (Gu et al., 2020; Orvieto et al., 2023). As a result, we employ the SSMs as our temporal-processing unit to meticulously balance the accuracy and efficiency of our CODEN framework.

### A.6.6 FORWARD PUSH

The prevailing paradigm for existing PPR-based propagation is derived from the concept of *Forward Push*, which calculates the approximated solution $\mathbf{z}_i = \sum_{l=0}^{\infty} \alpha(1 - \alpha)^l \left( \mathbf{A}^{(t)\top} \mathbf{D}^{(t)^{-1}} \right)^l \mathbf{x}_i^{(t)}$ ($1 \leq i \leq F$) under a given error bound $\epsilon$, where $\alpha$ is the decay factor of random walk. Specifically, *Forward Push* (depicted in Alg. 1) assigns the attribute vector $\mathbf{x}_i^{(t)}$ to the *residue vector* $\mathbf{r}^{(t)}$ (e.g., $\mathbf{r}^{(t)} = \mathbf{x}_i^{(t)}$), which represents the unpropagated mass of attribute vector $\mathbf{x}_i^{(t)}$. [6] We iteratively conduct the following two steps: (1) For each node $s \in \mathcal{V}^{(t)}$ such that $\mathbf{r}^{(t)}(s) > \epsilon$, $(1 - \alpha)$ fraction of $\mathbf{r}^{(t)}(s)$ will be propagated into the out-neighbors $t \in \mathcal{N}_{out}(s)$ averagely. (2) $(1 - \alpha)$ fraction of $\mathbf{r}^{(t)}(s)$ will be transferred into the reserve vector $\mathbf{h}^{(t)}$ and then $\mathbf{r}^{(t)}(s)$ is set as 0. This iteration will be terminated until $\mathbf{r}^{(t)}(s) \leq \epsilon$ for all nodes $s \in \mathcal{V}^{(t)}$ and we can deploy $\mathbf{h}^{(t)}$ as the approximated node embedding which satisfies $|\mathbf{h}^{(t)}(s) - \mathbf{z}^{(t)}(s)| \leq \epsilon$ for each node $s \in \mathcal{V}^{(t)}$.

### A.7 INTRODUCTION OF BASELINE METHODS

### A.7.1 SINGLE-SNAPSHOT METHODS

**Instant (Zheng et al., 2022), DynAnom (Guo et al., 2022), IDOL (Zhu et al., 2024).** These three methods inherit the updating skeleton of PPR (Zhang et al., 2016) to incrementally update the node

---

[6]In the following sections, we will omit the subscript $i$ for simplicity.

---

**Algorithm 1:** Forward Push

---

**Input** : Graph $\mathcal{G}^{(t)} = (\mathcal{V}^{(t)}, \mathcal{E}^{(t)})$, reserve vector $\boldsymbol{h}^{(t)}$, residue vector $\boldsymbol{r}^{(t)}$.
**Output**: Reserve vector $\boldsymbol{h}^{(t)}$

1   $\boldsymbol{h}^{(t)} = \boldsymbol{h}^{(t-1)}$;
2   **while** *exists* $s \in \mathcal{V}$ *such that* $\boldsymbol{r}^{(t)}(s) > \epsilon$ **do**
3      **foreach** $v \in \mathcal{N}_{out}^{(t)}(s)$ **do**
4         $\boldsymbol{r}^{(t)}(v) += (1 - \alpha) \cdot \dfrac{\boldsymbol{r}^{(t)}(s)}{|\mathcal{N}_{out}^{(t)}(s)|}$;
5      $\boldsymbol{h}^{(t)}(s) += \alpha \cdot \boldsymbol{r}^{(t)}(s); \boldsymbol{r}^{(t)}(s) = 0$;

---

embeddings. Following the analysis of (Zheng et al., 2022), the dominant complexity of updating embeddings can be formulated as $O\left(\frac{\|\boldsymbol{x}^{(t)}\|_1}{\epsilon n}\right)$ for a feature vector $\boldsymbol{x}^{(t)}$ assuming $\alpha$ as the constant. Hence we summarize the complexity of this catogory as $O\left(p \sum_{i=1}^{F} \|\boldsymbol{x}_i^{(t)}\|_1\right)$ for $F$ dimensions given $\epsilon = \frac{1}{n}$.

### A.7.2 RNN-BASED METHODS

**TGN (Rossi et al., 2020).** TGN proposes a general framework for learning the representation in CTDG, which models the past interactions between nodes using a compressed node state vector. Based on existing techniques, TGN provides flexible modules to compute the embeddings and update node states. For example, TGN utilizes the attention mechanism and GCN (Kipf & Welling, 2017) to compute the node embeddings and LSTM or GRU module to update the node state.

**TGCN (Zhao et al., 2019), EvolveGCN (Pareja et al., 2020), MPNN (Panagopoulos et al., 2021), ROLAND (You et al., 2022).** These four methods employ a unified pipeline to update the node embeddings and states. Specifically, these methods conduct the graph propagation based on GCN (Kipf & Welling, 2017) and update the state with GRU or LSTM units. Since each update will require to compute the embeddings from scratch, we summarize their complexity as $O(Km^{(t)}F)$ and $O(pn^{(t)}F^2)$ given $p$ updates.

### A.7.3 ATTENTION-BASED METHODS

**DySat (Sankar et al., 2020), ASTGCN (Guo et al., 2019).** DySat and ASTGCN utilize structural and temporal attention mechanism for dynamic graph representation learning. Both of them employ graph propagation by GCN. As a result, the graph propagation and state update require $O(n^2 F)$ time complexity for each update.

**TGAT (Xu et al., 2020).** TGAT adopts the GraphSage (Hamilton et al., 2017) for the embedding computation. Different from DySat, TGAT aims to formulate the interactions between the time encoding and node features. However, due to the polynomial property of the attention mechanism, TGAT still needs to consume $O(n^2 F)$ time for state update.

**DNNTSP (Yu et al., 2020), SEIGN(Qin et al., 2023), DyGFormer (Yu et al., 2023b).** These three methods follow a similar intuition to our alternative model CODEN-A, which intends to calculate the interactions between different time steps for each node. Given the time step $t$, this framework needs to consume $O(tn^2 F)$ time for the state update.

## A.8 ACCURACY-GUARANTEED EMBEDDING UPDATE

As we mentioned, our objective is to efficiently propagate frequent updates into the node embeddings from the preceding time step. The rationale behind this objective is that not all node embeddings can be significantly influenced by the updates, since an edge update causes only a minimal impact on distant nodes. However, current state-of-the-art algorithms (Zhou et al., 2022; Rossi et al., 2020; Li et al., 2023a) intricately intertwine graph convolution with node states utilizing trainable parameters in the message function MSG($\cdot$) of Equ. 1, requiring either repetitive convolution operations or a complete recalculation of states upon updates. To address these issues, we propose to incrementally

---

**Algorithm 2:** Embedding Update

**Input** : Graph $\mathcal{G}^{(t)} = (\mathcal{V}^{(t)}, \mathcal{E}^{(t)})$, update events $\Gamma = \{(u_1, v_1, t_1), (u_2, v_2, t_2), ...., (u_p, v_p, t_p)\}$,
reserve vector $\boldsymbol{h}^{(t)}$, old adjacency matrix $\boldsymbol{P}^{(t)}$, updated adjacency matrix $\boldsymbol{P}^{(t+p)}$.

**Output** : Reserve vector $\boldsymbol{h}^{(t+p)}$

1   $\boldsymbol{r}^{(t+p)} = \boldsymbol{0}$;

2   $\mathcal{V}_{changed} \leftarrow \{u|$ the nodes whose out degree has changed$\}$;

3   **foreach** $u \in \mathcal{V}_{changed}$ **do**

4      $\mathcal{N}_{add}(u) \leftarrow \{v|$ the added neighbors of $u\}$;

5      $\mathcal{N}_{del}(u) \leftarrow \{v|$ the deleted neighbors of $u\}$;

6      **foreach** $v \in \mathcal{N}_{add}(u)$ **do**

7         $\boldsymbol{r}^{(t+p)}(v) + = (1-\alpha)\boldsymbol{h}^{(t)}(u)/|\mathcal{N}_{out}^{(t+p)}(u)|$;

8      **foreach** $v \in \mathcal{N}_{del}(u)$ **do**

9         $\boldsymbol{r}^{(t+p)}(v) - = (1-\alpha)\boldsymbol{h}^{(t)}(u)/|\mathcal{N}_{out}^{(t)}(u)|$;

10      **foreach** $w \in \mathcal{N}_{out}^{(t+p)}(u) \setminus (\mathcal{N}_{add}(u) \cup \mathcal{N}_{del}(u))$ **do**

11         $\boldsymbol{r}^{(t+p)}(w) + = (1-\alpha)\boldsymbol{h}^{(t)}(u)\left(\frac{1}{|\mathcal{N}_{out}^{(t+p)}(u)|} - \frac{1}{|\mathcal{N}_{out}^{(t)}(u)|}\right)$;

12   *Forward Push* $(\mathcal{G}^{(t+p)}, \boldsymbol{h}^{(t)}, \boldsymbol{r}^{(t+p)})$;

---

integrate update events into the node embeddings $\boldsymbol{h}^{(t)}$ in the subsequent sections. This step will then trigger an update to the node states $\boldsymbol{M}^{(t)}(u)$ to effectively document the ongoing evolutionary process.

**Compensated Propagation.** In this section, we discuss how to efficiently update the node embedding $\boldsymbol{H}^{(t)}$ and let $||\boldsymbol{H}^{(t)} - \boldsymbol{Z}^{(t)}||_1 \leq n^{(t)}\epsilon$ holds for each time step $t$ as demonstrated in 1. Despite of the recent advancements to incrementally update the PPR-based embeddings (Zheng et al., 2022; Guo et al., 2022), it is pointed out that this invariant-based adjustment can not provide the accuracy bound after updating (Yoon et al., 2018a;b). To address this challenge, we draw on insights from recent works on Random Walk with Restart (RWR) (Yoon et al., 2018a) to compensate for embedding distance across different time steps.

Without loss of generality, we start with an example that $\boldsymbol{x}^{(t+1)} = \boldsymbol{x}^{(t)} \in \boldsymbol{X}^{(t)}$ and a new edge $e_{t+1} = (u, v)$ is added at time $t + 1$, creating an unbounded value difference between the old embedding $\boldsymbol{h}^{(t)}$ and the approximation target $\boldsymbol{z}^{(t+1)}$. Note that $\boldsymbol{h}^{(t)}$ is the approximation of $\boldsymbol{z}^{(t)}$, and we have:

$$
\begin{aligned}
\boldsymbol{z}^{(t+1)} - \boldsymbol{z}^{(t)} &= \alpha \left( \left( \boldsymbol{I} - (1-\alpha)\boldsymbol{P}^{(t+1)} \right)^{-1} - \left( \boldsymbol{I} - (1-\alpha)\boldsymbol{P}^{(t)} \right)^{-1} \right) \boldsymbol{x}^{(t+1)} \\
&\stackrel{(1)}{=} \alpha(1-\alpha) \left( \boldsymbol{I} - (1-\alpha)\boldsymbol{P}^{(t+1)} \right)^{-1} \Delta\boldsymbol{P} \left( \boldsymbol{I} - (1-\alpha)\boldsymbol{P}^{(t)} \right)^{-1} \boldsymbol{x}^{(t+1)} \\
&= \underbrace{\alpha \left( \boldsymbol{I} - (1-\alpha)\boldsymbol{P}^{(t+1)} \right)^{-1}}_{\text{propagation process}} \cdot \underbrace{(1-\alpha)\Delta\boldsymbol{P}\boldsymbol{z}^{(t)}}_{\text{compensated vector}}
\end{aligned} \tag{13}
$$

where (1) holds since $\boldsymbol{U}^{-1} - \boldsymbol{V}^{-1} = \boldsymbol{U}^{-1}(\boldsymbol{V} - \boldsymbol{U})\boldsymbol{V}^{-1}$ and $\Delta\boldsymbol{P} = \left( \boldsymbol{P}^{(t+1)} - \boldsymbol{P}^{(t)} \right)$. Based on Equ. 13, we have a key observation: *the difference between node embedding $\boldsymbol{z}^{(t)}$ and $\boldsymbol{z}^{(t+1)}$ can be compensated by propagating the new feature vector $(1-\alpha)\left( \boldsymbol{P}^{(t+1)} - \boldsymbol{P}^{(t)} \right)\boldsymbol{z}^{(t)}$ along the updated graph $\mathcal{G}^{(t+1)}$.* Since $\boldsymbol{h}^{(t)}$ is the approximation of $\boldsymbol{z}^{(t)}$ with the allowed error, we hence propose to implement the embedding update based on the *compensated vector* $(1-\alpha)\Delta\boldsymbol{P}\boldsymbol{h}^{(t)}$.

**Scalable Batch Update.** By adhering to the principle above, we extend the incremental update of node embedding into the batch setting, which is depicted in Alg. 2. Consider the graph $\mathcal{G}^{(t)}$ at time $t$ and the batch update events in $\Gamma = \{(u_1, v_1, t_1), (u_2, v_2, t_2), ...., (u_p, v_p, t_p)\}$ containing $p$ edge updates, where $(u_i, v_i)$ $(1 \leq i \leq p)$ will be viewed as deletion if it exists in $\mathcal{G}^{(t+i-1)}$ otherwise as an addition. First, we obtain the set $\mathcal{V}_{changed}$ denoting the nodes whose out-degree has changed. Given a node $u \in \mathcal{V}_{changed}$ and for each node $v \in \mathcal{N}_{out}^{(t)}(u) \cup \mathcal{N}_{out}^{(t+p)}(u)$, the corresponding entry of

---

**Algorithm 3:** CODEN on Continous Prediction

---

**Input** :Initial Graph $\mathcal{G}^{(0)} = (\mathcal{V}^{(0)}, \mathcal{E}^{(0)})$, initial embedding $\boldsymbol{H}^{(0)}$, update events
$\{(u_1, v_1, t_1), (u_2, v_2, t_2), ...., (u_k, v_k, t_k), ...\}$, corresponding prediction time
$T = \{t_1, t_2, ...., t_k, ...\}$.

**Output** :Prediction result $\boldsymbol{Y}^{(t_{k+1})}$ for $k = 0, 1, 2, 3, ...$

1 $\sigma = 0; \Gamma = \emptyset; t = 0; k = 0$ ;

2 **for** $k = 0, 1, 2, 3, ...$ **do**

3     **foreach** *update $(u_t, v_t)$ between $t_k$ and $t_{k+1}$* **do**

4        $\sigma+ = \frac{1-\alpha}{\alpha} \left\| \left( \boldsymbol{P}^{(t)} - \boldsymbol{P}^{(t-1)} \right) \cdot \boldsymbol{x}_{max} \right\|_1 + 2n\epsilon$;

5        $\Gamma.\text{add}((u_t, v_t, t))$;

6        **if** $\sigma > \lambda$ *or* $t = t_{k+1}$ **then**

7           **foreach** $h^{(t-|\Gamma|)} \in \boldsymbol{H}^{(t-|\Gamma|)}$ *in parallel* **do**

8              *Embedding Update*$(\mathcal{G}^{(t-|\Gamma|)}, \Gamma, h^{(t-|\Gamma|)}, \boldsymbol{P}^{(t-|\Gamma|)})$,

9              $\boldsymbol{P}^{(t)}); \sigma = 0; \Gamma = \emptyset$;

10           $\boldsymbol{M}^{(t)} = \bar{\mathcal{A}} \cdot \boldsymbol{M}^{(t-|\Gamma|)} + \bar{\mathcal{B}} \cdot \boldsymbol{H}^{(t)}$;

11     $\boldsymbol{Y}^{(t_{k+1})} = \text{MLP}(\boldsymbol{M}^{(t_{k+1})})$;

---

$\Delta \boldsymbol{P} = \boldsymbol{P}^{(t+p)} - \boldsymbol{P}^{(t)}$ can be expressed as:

$$
\Delta \boldsymbol{P}[v, u] = \begin{cases} \frac{1}{|\mathcal{N}_{out}^{(t+p)}(u)|}, & v \text{ is the added neighbor of } u, \\ -\frac{1}{|\mathcal{N}_{out}^{(t)}(u)|}, & v \text{ is the deteted neighbor of } u, \\ \frac{1}{|\mathcal{N}_{out}^{(t+p)}(u)|} - \frac{1}{|\mathcal{N}_{out}^{(t)}(u)|}, & \text{otherwise.} \end{cases}
$$

Then, we multiply $\Delta \boldsymbol{P}[v, u]$ with $(1 - \alpha)\boldsymbol{h}^{(t)}(u)$ and finish the computation of the compensated vector $(1 - \alpha) \left( \boldsymbol{P}^{(t+p)} - \boldsymbol{P}^{(t)} \right) \boldsymbol{h}^{(t)}$ (lines 6-11). Moreover, we assign this new feature vector as the unpropagated residue vector $\boldsymbol{r}^{(t+p)}$. Note that $\boldsymbol{r}^{(t+p)}$ may exceed the permissible error for certain nodes, such as when $\boldsymbol{r}^{(t+p)}(u) > \epsilon$ ($u \in \mathcal{V}^{(t+p)}$). Consequently, we trigger the *Forward Push* mechanism (Andersen et al., 2006) again to propagate the residue vector $\boldsymbol{r}^{(t+p)}$ (line 12), which will subsequently influence the embedding values of other nodes and ensure the desired accuracy.

**Theoretical Accuracy Guarantee.** Alg. 2 returns the updated embeddings by propagating the compensated vector along the updated graph, where the compensated vector is derived using $\boldsymbol{z}^{(t)}$, as specified in Equ. 13. Although we process this vector with the approximated vector $\boldsymbol{h}^{(t)}$, we can still establish an error bound for the output vector $\boldsymbol{h}^{(t+p)}$, which is the result of propagation from scratch at time $t + p$. We formally state this accuracy guarantee in the following lemma:

**Lemma 3.** *Given the normalized adjacency matrix $\boldsymbol{P}^{(t)}$ at time $t$ and the update event set $\Gamma = \{(u_1, v_1), (u_2, v_2), ...., (u_p, v_p)\}$, Alg 2 can output the approximated embedding vector $\boldsymbol{h}^{(t+p)}$ which satisfies:*

$$
||\boldsymbol{H}^{(t+p)} - \boldsymbol{Z}^{(t+p)}||_1 \leq n^{(t)}\epsilon
$$

*where $\boldsymbol{Z}^{(t+p)} = \sum_{l=0}^{\infty} \alpha(1 - \alpha)^l \left( \boldsymbol{P}^{(t+p)} \right)^l \boldsymbol{X}^{(t+p)}$.*

*Proof.* According to (Zheng et al., 2022), the estimated vector $\hat{h}^{(t-1)}$ at time $t - 1$ and the exact PPR vector $\sum_{l=0}^{\infty} \alpha \left( (1 - \alpha)^l \boldsymbol{P}^{(t-1)} \right)^l \cdot \boldsymbol{x}_i$ have the following relationship:

$$
\sum_{l=0}^{\infty} \alpha \left( (1 - \alpha)^l \boldsymbol{P}^{(t-1)} \right)^l \boldsymbol{x}_i = \hat{h}^{(t-1)} + \sum_{l=0}^{\infty} \alpha \left( (1 - \alpha)^l \boldsymbol{P}^{(t-1)} \right)^l \boldsymbol{r}_i
$$

$$
= \hat{h}^{(t-1)} + \alpha \left( I - (1 - \alpha)\boldsymbol{P}^{(t-1)} \right)^{-1} \boldsymbol{r}_i
$$

For the purpose of clear demonstration, we express $(1-\alpha)P^{(t)}$ as $P^{(t)}$ for the following proof. Then we formulate the output $\hat{h}^{(t)}$ of Alg. 2 as:

$$
\hat{h}^{(t)} = \sum_{j=0}^{\infty}(P^{(t)})^j(P^{(t)} - P^{(t-1)})\,(\alpha\sum_{i=0}^{\infty}(P^{(t-1)})^i\boldsymbol{x}_i - \alpha\sum_{i=0}^{\infty}(P^{(t-1)})^i
$$

$$
\boldsymbol{r}_i^{(t-1)}) + \hat{h}^{(t-1)}
$$

$$
= \sum_{j=0}^{\infty}(P^{(t)})^j(P^{(t)} - P^{(t-1)})\alpha\sum_{i=0}^{\infty}(P^{(t-1)})^i\boldsymbol{x}_i - \sum_{j=0}^{\infty}(P^{(t)})^j(P^{(t)}-
$$

$$
P^{(t-1)})\alpha\sum_{i=0}^{\infty}(P^{(t-1)})^i\boldsymbol{r}_i^{(t-1)} + \hat{h}^{(t-1)} \tag{14}
$$

The first term of the last line in Equ. 14 can be simplified as:

$$
\sum_{j=0}^{\infty}(P^{(t)})^j(P^{(t)} - P^{(t-1)})\alpha\sum_{i=0}^{\infty}(P^{(t-1)})^i\boldsymbol{x}_i
$$

$$
= \sum_{j=1}^{\infty}(P^{(t)})^j\alpha\sum_{i=0}^{\infty}(P^{(t-1)})^i\boldsymbol{x}_i - \sum_{j=0}^{\infty}(P^{(t)})^j\alpha\sum_{i=j}^{\infty}(P^{(t-1)})^i\boldsymbol{x}_i
$$

$$
= \sum_{j=1}^{\infty}(P^{(t)})^j\alpha\left(\sum_{i=1}^{\infty}(P^{(t-1)})^i + \boldsymbol{I}\right)\boldsymbol{x}_i - \left(\sum_{j=1}^{\infty}(P^{(t)})^j + \boldsymbol{I}\right)\alpha\sum_{i=1}^{\infty}(P^{(t-1)})^i\boldsymbol{x}_i
$$

$$
= \alpha\sum_{j=1}^{\infty}(P^{(t)})^i\boldsymbol{x}_i - \alpha\sum_{i=1}^{\infty}(P^{(t-1)})^i\boldsymbol{x}_i = \alpha\sum_{j=0}^{\infty}(P^{(t)})^i\boldsymbol{x}_i - \alpha\sum_{i=0}^{\infty}(P^{(t-1)})^i\boldsymbol{x}_i
$$

Similarly, the second term can be formed as:

$$
\sum_{j=0}^{\infty}(P^{(t)})^j(P^{(t)} - P^{(t-1)})\alpha\sum_{i=0}^{\infty}(P^{(t-1)})^i\boldsymbol{r}_i^{(t-1)}
$$

$$
= \alpha\sum_{j=0}^{\infty}(P^{(t)})^i\boldsymbol{r}_i^{(t-1)} - \alpha\sum_{i=0}^{\infty}(P^{(t-1)})^i\boldsymbol{r}_i^{(t-1)}
$$

Then Equ. 14 is further expressed as:

$$
\hat{h}^{(t)} = \alpha\sum_{j=0}^{\infty}(P^{(t)})^i\boldsymbol{x}_i - \alpha\sum_{i=0}^{\infty}(P^{(t-1)})^i\boldsymbol{x}_i - \alpha\sum_{j=0}^{\infty}(P^{(t)})^i\boldsymbol{r}_i^{(t-1)}+
$$

$$
\alpha\sum_{i=0}^{\infty}(P^{(t-1)})^i\boldsymbol{r}_i^{(t-1)} + \alpha\sum_{i=0}^{\infty}(P^{(t-1)})^i\boldsymbol{x}_i - \alpha\sum_{i=0}^{\infty}(P^{(t-1)})^i\boldsymbol{r}_i^{(t-1)}
$$

$$
= \alpha\sum_{j=0}^{\infty}(P^{(t)})^i\boldsymbol{x}_i - \alpha\sum_{j=0}^{\infty}(P^{(t)})^i\boldsymbol{r}_i^{(t-1)}
$$

Since the $\boldsymbol{r}_i^{(t-1)}(u) < \epsilon$ for each $u \in \mathcal{V}^{(t-1)}$, the proof is finished.

$\square$

**Comparison with the invariant-based scheme.** Compared with the invariant-based scheme in existing related works (Zheng et al., 2022; Guo et al., 2022; Zhu et al., 2024), we claim that our method stands out as the first to offer guaranteed accuracy of updated embeddings in dynamic settings. The rationale for this difference is that our method offers a more stable adjustment by leveraging a compensate vector, making it more robust in scenarios with significant node degree changes. A running example underpinning this insight is further provided in the Appendix. In summary, our accuracy-guaranteed embedding update forms the novel theoretical foundation of CODEN (e.g., Lemma 1 and Proposition 1), thereby conferring substantial benefits in producing high-quality representations.

### A.9 EXTENSION TO ATTRIBUTE CHANGES.

Note that we have made an assumption that $\boldsymbol{x}^{(t+p)} = \boldsymbol{x}^{(t)}$ such that we can focus on the topological change of the graph. Actually, this assumption can be easily relaxed since extending our approach to

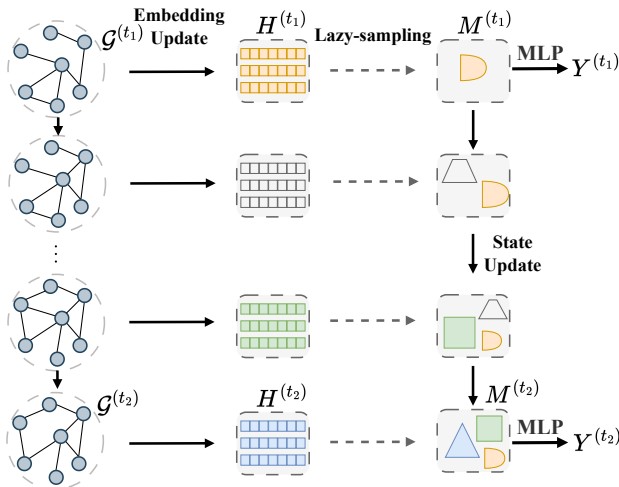

Figure 6: An illustration of the CODEN framework. $\boldsymbol{M}^{(t)}$, $\boldsymbol{H}^{(t)}$ and $\boldsymbol{Y}^{(t)}$ denote the node state matrix, embedding matrix and the prediction results at time $t$, respectively.

dynamic-attribute graphs is much more straightforward. Specifically, given $\Delta\boldsymbol{x} = \boldsymbol{x}^{(t+p)} - \boldsymbol{x}^{(t)}$ and $\boldsymbol{P}^{(t+p)} = \boldsymbol{P}^{(t)}$, we can naturally derive the difference between $\boldsymbol{z}^{(t)}$ and $\boldsymbol{z}^{(t+p)}$ as:

$$\boldsymbol{z}^{(t+p)} - \boldsymbol{z}^{(t)} = \alpha \left( I - (1-\alpha)\boldsymbol{P}^{(t+1)} \right)^{-1} \Delta\boldsymbol{x}. \tag{15}$$

Therefore, we can view $\Delta\boldsymbol{x}$ as the compensated vector and easily obtain the updated embedding by directly invoking *Forward Push* $(\mathcal{G}^{(t+p)}, \boldsymbol{h}^{(t)}, \Delta\boldsymbol{x})$. [7]

### A.10 COMPLEXITY ANALYSIS AND DISCUSSION BETWEEN CODEN-A AND CODEN

In this section, we analyze the time complexity of CODEN, which is dominated by two processes: the operation in the initial propagation, the complexity of updating node embeddings, and the complexity of updating node states. We will analyze the complexity under two edge arrival patterns: (i) *the average case:* edges arriving randomly, with each node having an equal probability of being the starting node, and (ii) *the worst case:* edges arriving randomly, but with varying probabilities for each node to be the starting node.

**Complexity of updating node embeddings.** Given the attribute vector $\boldsymbol{x}^{(0)}$, the complexity for propagating this vector using *Forward Push* with error $\epsilon$ can be bounded by $O(\|\boldsymbol{x}^{(0)}\|_1/\alpha\epsilon)$ according to (Liao et al., 2022; Andersen et al., 2006). We show our results of complexity analysis as follows:

**Lemma 4.** *Assuming there exist $p$ edge updates during the prediction time $t_k$ and $t_{k+1}$, the complexity of updating node embeddings from $t_k$ to $t_{k+1}$ is $O\left( \frac{p}{\epsilon n^{(t_k)}} \sum_{i=1}^{F} \|\boldsymbol{x}_i^{(t_k)}\|_1 \right)$ under pattern (i) and $O\left( \frac{p}{\epsilon} \sum_{i=1}^{F} \max_{u \in \mathcal{V}^{(t)}} |\boldsymbol{x}_i^{(t_k)}(u)| \right)$ under pattern (ii).*

*Proof.* We first discuss the simple situation where only one edge $(u, v)$ is added into graph $\mathcal{G}^{(t)}$ and transfer it into $\mathcal{G}^{(t+1)}$. Then we can naturally derive the propagation complexity of a single dimension as $\|(1-\alpha) \left( \boldsymbol{P}^{(t+1)} - \boldsymbol{P}^{(t)} \right) \boldsymbol{h}^{(t)}\|_1/\alpha\epsilon$, where $\boldsymbol{h}^{(t)}$ is one dimension of $\boldsymbol{H}^{(t)}$. As demonstrated in

---

[7]In the remaining sections, we mainly focus on the topology change and assume $\boldsymbol{X}^{(t)}$ remains unchanged in the following sections for a clear presentation.

Alg. 2, given an added edge $(u, v)$ the residue of $\|(1 - \alpha) \left( \boldsymbol{P}^{(t+1)} - \boldsymbol{P}^{(t)} \right) \boldsymbol{h}^{(t)}\|_1$ can be divided as:

$$\|(1 - \alpha) \left( \boldsymbol{P}^{(t+1)} - \boldsymbol{P}^{(t)} \right) \boldsymbol{h}^{(t)}\|_1 = \left| (1 - \alpha) \boldsymbol{h}^{(t)}(u) / |\mathcal{N}_{out}^{(t+1)}(u)| \right|$$

$$+ \left| \sum_{w \in \mathcal{N}_{out}^{(t+1)}(u)} (1 - \alpha) \boldsymbol{h}^{(t)}(u) \left( 1/|\mathcal{N}_{out}^{(t+1)}(u)| - 1/|\mathcal{N}_{out}^{(t)}(u)| \right) \right|$$

$$\leq (1 - \alpha)|\boldsymbol{h}^{(t)}(u)| + \left| |\mathcal{N}_{out}^{(t+1)}(u)| \cdot (1 - \alpha) \boldsymbol{h}^{(t)}(u) \cdot 1/|\mathcal{N}_{out}^{(t+1)}(u)| \right|$$

$$\overset{(1)}{\leq} 2(1 - \alpha)|\boldsymbol{\phi}^{(t)}(u)| + 2(1 - \alpha)\epsilon,$$

where $\boldsymbol{\phi}^{(t)} = \alpha \left( I - (1 - \alpha) \boldsymbol{P}^{(t)} \right)^{-1} \boldsymbol{x}^{(t)}$ and (1) holds since $\boldsymbol{h}^{(t)}(u)$ is the underestimation of $\boldsymbol{\phi}^{(t)}(u)$. Under pattern (i), edges arrive in the system randomly so that each node has the same probability of being the start point of the changed edge. Therefore we can obtain the expected time for each update as $E\left[ \frac{2(1-\alpha)|\boldsymbol{\phi}^{(t)}(u)| + 2(1-\alpha)\epsilon}{\alpha\epsilon} \right] = \frac{2(1-\alpha)\|\boldsymbol{x}^{(t)}\|_1}{\epsilon n^{(t)}} + \frac{2(1-\alpha)}{\alpha}$. Given there exist $p$ edge updates in a batch at time $t_k$, we add the superscript of $\boldsymbol{x}^{(t_k)}$ and formulate the complexity for propagation as $O(p \sum_{i=1}^{F} \frac{\|\boldsymbol{x}_i^{(t_k)}\|_1}{\epsilon n^{(t_k)}})$. Under pattern (ii), we have $\frac{2(1-\alpha)|\boldsymbol{\phi}^{(t)}(u)|}{\alpha\epsilon} \leq \frac{2(1-\alpha) \max_{u \in \mathcal{V}} |\boldsymbol{x}^{(t)}(u)|}{\epsilon}$, which indicates the complexity of this case is $O(p \sum_{i=1}^{F} \frac{\max_{u \in \mathcal{V}} |\boldsymbol{x}_i^{(t_k)}(u)|}{\epsilon})$ ☐

**Complexity of updating node states.** Since the number of sampled embeddings can affect the time of training in SSM, hence we first solve the problem of how many embeddings will be sampled given $p$ update events and the threshold $\lambda$. Then we can compute the number of iterations and the complexity of updating node states. We formally show the result with the following lemma:

**Lemma 5.** *Updating states requires* $O\left( \frac{pF^2}{\lambda}(\|\boldsymbol{x}_{max}\|_1 + \epsilon(n^{(t)})^2) \right)$ *time under pattern (i) and* $O\left( \frac{pF^2 n^{(t)}}{\lambda} \max_{u \in \mathcal{V}^{(t)}}(|\boldsymbol{x}_{max}(u)| + \epsilon n^{(t)}) \right)$ *under pattern (ii), where* $\{\boldsymbol{x}_{max}\}_i = \max_{1 \leq j \leq F} |\boldsymbol{X}_{ij}^{(t)}|$.

*Proof.* Based on Alg. 3, for an edge update $(u, v)$, the increased mass of $\sigma$ is at most $\frac{1-\alpha}{\alpha} \left\| \left( \boldsymbol{P}^{(t)} - \boldsymbol{P}^{(t-1)} \right) \cdot \boldsymbol{x}_{max} \right\|_1 + 2n\epsilon$. Following the derivation of Lemma 4, we can naturally obtain the upper bound of this term as $2(1 - \alpha)|\boldsymbol{x}_{max}(u)| + 2n\epsilon$. Therefore, the largest number of edges contained before $\sigma > \lambda$ is at least $\lambda / (2(1 - \alpha)|\boldsymbol{x}_{max}(u)| + 2n\epsilon)$. Then given $p$ update events, the number of embeddings sampled $L$ can be bounded as $L \leq (2(1 - \alpha)|\boldsymbol{x}_{max}(u)| + 2n\epsilon)p/\lambda$. Finally, the time complexity of updating the node state in SSM is $O(n^{(t)} L F^2)$ (Li et al., 2024a; Gu & Dao, 2023). Therefore, we can naturally obtain the complexity of updating node states as $O(\frac{\|\boldsymbol{x}_{max}\|_1 + \epsilon\left(n^{(t)}\right)^2}{\lambda} pF^2)$ under pattern (i) and $O(\frac{\max_{u \in \mathcal{V}^{(t)}} |\boldsymbol{x}_{max}(u)|n^{(t)} + \epsilon\left(n^{(t)}\right)^2}{\lambda} pF^2)$ under pattern (ii). ☐

**Discussion.** As shown in Tab. 1, CODEN remarkably achieves superior complexity in the continuous prediction for TGNNs. To thoroughly examine the rationale underlying its property, we provide a comprehensive comparison with other three categories of TGNNs as following:

• **Efficient update as single-snapshot methods.** CODEN simplified the traditional graph convolution such as GCN (Kipf & Welling, 2017) and directly utilizes the parameter-free node embedding as the basement. This design avoid the common $O(Km^{(t)}F)$ time required for graph convolution process on the updated graph, which efficiently facilitate the subsequent update of node state.

• **Optimized complexity for RNN-based methods.** We observe that CODEN aligns with the iterative framework of RNN-based methods, which typically incur a complexity $O(pn^{(t)}F^2)$ for node state update and parameter training. Additionally, CODEN employs a lazy-sampling strategy to manage the sequence length, significantly enhancing the efficiency of iteration. This efficiency can be controlled by adjusting the parameter $\lambda$. Specifically, when $\lambda$ is small (e.g., $\lambda = O(1)$), CODEN processes each update individually, resulting in a runtime comparable to that of RNN-based methods.

• **Effectiveness as Attention-based methods.** The commonly used attention mechanism costs $O(p\left(n^{(t)}\right)^2 F)$ to achieve enhanced temporal memorization. As discussed previously, we success-

fully establish a theoretical connection between CODEN and this mechanism, uncovering the rationale behind CODEN's effectiveness. This critical insight allows CODEN to achieve comparable accuracy while significantly reducing computational complexity.

**Comparison between** CODEN-A **and** CODEN. Next, we will compare CODEN and CODEN-A with respect to the time complexity and the representation quality:

**Proposition 4.** *Compared with* CODEN-*A,* CODEN *not only reduces time complexity but also mitigates over-smoothing under the Dirichlet energy measure.*

*Proof.* (i) **Time complexity comparison.** Given the prediction time $t$, CODEN-A is required to calculate the dependence between the embedding at $t$ and other time steps. Formally, we obtain the state at time $t$ as: $\boldsymbol{M}^{(t)} = \sum_{s=0}^{t} \text{softmax}\left( \frac{(\boldsymbol{W}_q \boldsymbol{H}^{(t)}) \cdot (\boldsymbol{W}_k \boldsymbol{H}^{(s)})^\top}{\sqrt{F'}} \right) \cdot \left( \boldsymbol{W}_v \boldsymbol{H}^{(s)} \right)$. Compared with the complexity of CODEN ($O\left( \frac{\|\boldsymbol{x}_{max}\|_1 + n^{(t)}}{\lambda} p F^2 \right)$), CODEN-A suffers from a $O(t(n^{(t)})^2 F)$ time complexity when the time window is extended to $t$.

(ii) **Quality comparison based on Dirichlet Energy.** As an emerging measurement to access the quality of node representations, the degree of over-smoothing becomes pronounced as GNNs deepen and the node representations go indistinguishable, which significantly degrades the performance in the downstream tasks. To quantify the distance of node pairs, we employ the metric of *Dirichlet Energy* to observe how SSM can alleviate the over-smoothing issue compared with the attention mechanism. Given the state $\boldsymbol{M}^{(t)}$ of nodes, the Dirichlet Energy $\mathbb{DE}(\boldsymbol{M}^{(t)})$ is defined as follows:

$$\mathbb{DE}(\boldsymbol{M}^{(t)}) = \text{tr}\left( \boldsymbol{M}^{(t)}(\boldsymbol{I} - \boldsymbol{A}^{(t)^\top} \boldsymbol{D}^{(t)^{-1}})\boldsymbol{M}^{(t)^\top} \right) = \frac{1}{2}\sum \boldsymbol{A}_{ij}^{(t)}\|\frac{\boldsymbol{M}^{(t)}(i)}{\sqrt{1 + |\mathcal{N}_{out}^{(t)}(i)|}} - \frac{\boldsymbol{M}^{(t)}(j)}{\sqrt{1 + |\mathcal{N}_{out}^{(t)}(j)|}}\|_2^2$$

where $\text{tr}(\cdot)$ denotes the trace of the matrix and $\boldsymbol{A}_{ij}^{(t)}$ is the $(i, j)$-th element of $\boldsymbol{A}^{(t)}$. Dirichlet Energy reflects the smoothness of adjacent node representations. Therefore, a relatively larger value of $\mathbb{DE}(\boldsymbol{M}^{(t)})$ reveals that the representations of different nodes can be more distinct, making it easier to separate different nodes in classification tasks. Based on this insight, we investigate the potential over-smoothing issues of CODEN and CODEN-A by analyzing the Dirichlet Energy of their state matrices:

**Lemma 6.** *Denoting the state matrix of* CODEN *and* CODEN-*A as* $\boldsymbol{M}_C^{(t)}$ *and* $\boldsymbol{M}_A^{(t)}$ *respective, we have:* $\mathbb{DE}(\boldsymbol{M}_A^{(t)}) \leq \mathbb{DE}(\boldsymbol{M}_C^{(t)})$, *if* $F \cdot \sigma_{min}^2(\prod_{i=s+1}^{t} \bar{\mathcal{A}}^{(i)}\bar{\mathcal{B}}^{(s)}) \geq 1$ *for* $0 \leq s \leq t$, *where* $\sigma_{min}(\cdot)$ *means the minimum eigenvalue of the matrix.*

We collect all learnable transition matrices in $\mathcal{W} = \left\{ \bar{\mathcal{A}}^{(i)}, \bar{\mathcal{B}}^{(s)} \mid 1 \leq i \leq T, \ 0 \leq s \leq T-1 \right\}$, where $T$ is the overall length of the edge sequence. Following the soft-orthogonality regularizer of (Huang et al., 2018), we penalize every $W \in \mathcal{W}$ with $\mathcal{R}_{\text{orth}}(W) = \left\| W^\top W - \gamma^2 I \right\|_F^2, \gamma = c\,F^{-1/(2T)}(c > 1)$. This term pulls all singular values of parameters toward $\gamma$ therefore guarantees in practice $F \cdot \sigma_{\min}^2\left( \prod_{i=s+1}^{t} \bar{\mathcal{A}}^{(i)}\bar{\mathcal{B}}^{(s)} \right) \geq 1, 0 \leq s \leq t$ (see Lemma 2 for the detailed proof). $\square$

### A.11 DISCUSSION OF THE INVARIANT-BASED SCHEME

Considering $p$ edge updates, existing related works (Zheng et al., 2022; Guo et al., 2022; Zhu et al., 2024) adhere the invariant relationship in PPR and perform the following rules to locally update the node embedding: $\boldsymbol{r}^{(t+p)}(u) \leftarrow \boldsymbol{r}^{(t)}(u) - \frac{\Delta_p(u)\boldsymbol{h}^{(t)}(u)}{\alpha|\mathcal{N}_{out}^{(t)}(u)|}, \boldsymbol{r}^{(t+p)}(v) \leftarrow \boldsymbol{r}^{(t)}(v) + \frac{(1-\alpha)\Delta_p(u)\boldsymbol{h}^{(t)}(u)}{\alpha|\mathcal{N}_{out}^{(t)}(u)|}, \boldsymbol{h}^{(t+p)}(u) \leftarrow \boldsymbol{h}^{(t)}(u) \cdot \frac{|\mathcal{N}_{out}^{(t+p)}(u)|}{|\mathcal{N}_{out}^{(t)}(u)|}$, where $\Delta_p(u) = |\mathcal{N}_{out}^{(t+p)}(u)| - |\mathcal{N}_{out}^{(t)}(u)|$. The *Forward Push* process is then repeated to ensure that local adjustments propagate to distant nodes. Although these methods follow a similar pipeline to CODEN, we emphasize that our initialization process prior to *Forward Push* is fundamentally different.

To illustrate this distinction, we present a toy example in Fig. 7 where 3 edges are added between node $v_0$ and $v_2 \sim v_4$. Based on this result, we see that the invariant-based scheme can not guarantee the accuracy of node embeddings since $|\boldsymbol{h}(v) - \boldsymbol{z}(v)| = |0.75 - 0.08| > 0.5 = \epsilon$ for $v = v_2 \sim v_4$.

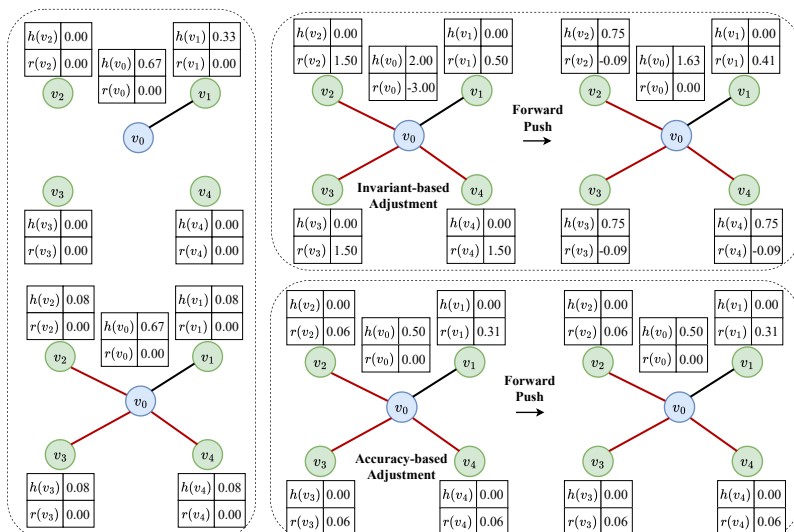

(a) *Propagation from the scratch* (b) *Result of variant-based and accuracy-based adjustment*

Figure 7: A toy example to compare the results of 4 kinds of push operations when we set $\alpha = 0.5$ and $\epsilon = 0.5$. (a) shows the initial push result before the update. (b) shows the ground truth result of the push from scratch. (c) shows the result of the variant-based push, where the error of the embeddings of node $v_2 \sim v_4$ have exceeded the bound $\epsilon$. (d) shows the result of the accuracy-based push, where the accuracy of node embeddings is guaranteed.

In contrast, our accuracy-based can still maintain the embedding accuracy after the *Forward Push* algorithm, which confers substantial benefits in producing high-quality representations for later node state formulation.

## A.12 ADDITIONAL EXPERIMENTAL RESULTS

### A.12.1 DATASETS

We adopt five representative real-world dynamic datasets: *DBLP* (Li et al., 2023a), *Tmall* (Lu et al., 2019), and three large-scale graphs, *Reddit* (Hamilton et al., 2017), *Patent* (Hall et al., 2001) and *Papers100M* (Hu et al., 2020). The statistics of the datasets are shown in Tab. 2. Within these datasets, the training, validation, and test sets are randomly allocated in proportions of 70%, 10%, and 20% respectively. To simulate scenarios that necessitate continuous prediction, we adopt the experimental framework outlined in (Zheng et al., 2022) and (Zhu et al., 2024), where the graph is segmented into an initial graph and $|T|$ batches of edge sequences. Then, each batch of edges will be added at distinct time steps in a dynamically evolving state. All methods are evaluated through a pipeline encompassing both training and inference.

Table 7: Parameter settings. Here "lr" means the learning rate, "$K$" means the number of convolution layers, "hidden" means the hidden size of the network, and "batch number" means the number of neighbor sampling for baselines.

| Datasets | lr | K | hidden | batch number |
|---|---|---|---|---|
| *DBLP* | 1e-3 | 4 | 1024 | 12 |
| *Tmall* | 1e-3 | 4 | 1024 | 12 |
| *Reddit* | 1e-3 | 4 | 512 | 12 |
| *Patent* | 1e-3 | 4 | 512 | 12 |
| *Papers100M* | 1e-3 | 2 | 512 | 12 |

### A.12.2 EXPERIMENTAL SETTING

Since most of the compared RNN-based and attention-based methods are designed for small-scale graphs, we adopt the neighboring sampling techniques (Hamilton et al., 2017) for these methods when

applied on the *Reddit* and *Patent* datasets to avoid the out-of-memory problem. We summarize the experimental settings for all baselines in Tab. 7, where the common parameters, such as learning rate and hidden size, are maintained consistently for CODEN. Specifically, we set the threshold $\epsilon = 1e^{-7}$, $\lambda = 0.1$, $F' = 16$, and $\alpha = 0.2$ by default in CODEN. All experimental results are obtained with 5 runs on a Linux machine with an Intel(R) Xeon(R) Gold 6238R CPU @ 2.20GHz with 160GB RAM and an NVIDIA RTX A5000 with 24GB memory. All results are averaged over 10 runs.

### A.12.3 MICRO-F1 SCORES AND TRAINING TIME ON DBLP AND TMALL

The additional Micro-F1 scores and training time on DBLP and Tmall datasets are presented in Fig. 8 and 9.

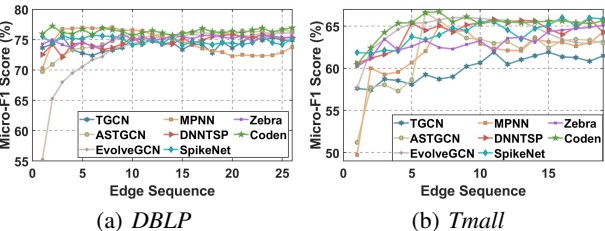

(a) *DBLP*       (b) *Tmall*

Figure 8: Micro-F1 scores for each prediction time.

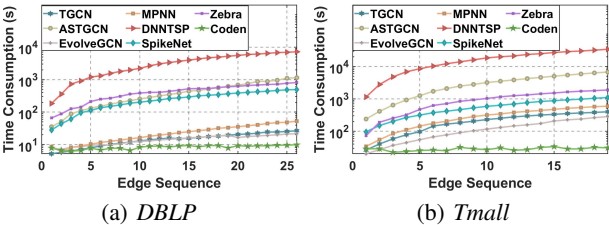

(a) *DBLP*       (b) *Tmall*

Figure 9: Training time consumption for each prediction time.

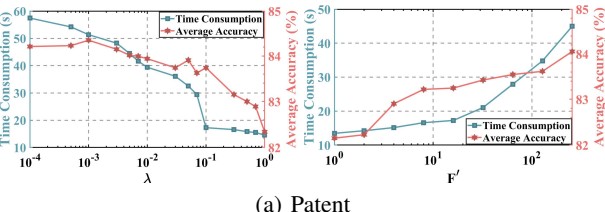

(a) Patent

Figure 10: The average training time and the average accuracy when setting different $\lambda$ and $F'$ on *Patent* dataset.

### A.12.4 INFORMATION COMPRESSION UNDER FLEXIBLE ARRIVAL PATTERNS.

In the experiments above, we added edges to the initial graph following an edge-inclined arrival pattern. However, using a more flexible arrival pattern would allow for a more thorough examination of the models' memory capabilities, particularly in assessing whether they experience significant performance degradation. Therefore, we employ more edge arrival patterns on each model and report the corresponding accuracy on *Reddit* dataset, as depicted in Fig. 11. In the edge-declined pattern, we reverse the order of the edge-inclined setting and remove the batch of edges sequentially from the final snapshot. Then, we randomly add and remove one batch of edges simultaneously to the initial graph, which is denoted as the edge-balanced pattern. In the edge-declined pattern, we observe that as the edges are missing gradually, all methods suffer from a performance decrease. Interestingly, at time step 16 under the edge-declined setting, where the graph remains identical to the initial graph used at time step 1 of the edge-inclined pattern (Fig. 1), CODEN achieves a notably higher accuracy (91.86% vs 87.86%), indicating its robustness of memory to edge removals. Moreover, CODEN

demonstrates a significant improvement in this pattern, achieving an average prediction accuracy of 93.48%, indicating its ability to effectively compress historical information for future predictions.

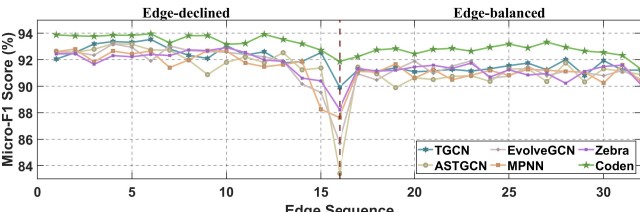

Figure 11: Micro-F1 scores under the Edge-declined and Edge-balanced patterns on *Reddit* dataset.

## A.13 LIMITATION AND FUTURE WORK

While CODEN demonstrates strong performance in terms of accuracy, efficiency, and scalability across various dynamic graph benchmarks, there are several limitations and open directions worth exploring.

First, CODEN has been primarily evaluated on homogeneous dynamic graphs. Its applicability to heterogeneous graphs, where nodes and edges may belong to different types and carry diverse attributes, remains an open question. Extending CODEN to heterogeneous temporal graphs would improve its versatility in real-world applications such as knowledge graphs or multi-modal social networks. Second, we have not explicitly incorporated uncertainty estimation in the prediction process. For high-stakes applications (e.g., traffic forecasting or financial modeling), extending CODEN with confidence-aware or Bayesian mechanisms could provide more trustworthy predictions.

In future work, we plan to address these limitations and investigate the use of CODEN in domains such as reinforcement learning on dynamic environments and human-robot interaction graphs.

## A.14 BROAD IMPACT

This work contributes to foundational understanding and methodology in machine learning, and is not tied to specific applications or deployments. As such, we do not foresee direct societal impacts—positive or negative—arising from the current research. While it may eventually support downstream applications, the work itself does not interface with sensitive domains, decision-making systems, or data related to privacy, security, or fairness.

