# OpenReview forum: "Continuous Temporal Graph Neural Networks"
_ICLR.cc/2026/Conference — Submitted to ICLR 2026_

### Official Review · Reviewer_uUcu · 2025-10-23

**Soundness:** 2
**Presentation:** 2
**Contribution:** 3
**Rating:** 4
**Confidence:** 3

**Summary:**

This paper aims to propose an efficient and effective TGNN model for link prediction that can adapt to dynamic graph changes. It proposes Coden, by introducing an efficient incremental update mechanism for node embeddings and leveraging State Space Models (SSMs) for state summarization, which effectively compresses historical information. Theoretical analysis demonstrates Coden's efficiency, and the empirical studies show its effectiveness.

**Strengths:**

S1. The proposed method, Coden, improves in both efficiency and effectiveness, compared to existing methods.

S2. It provides theoretical guarantees on the efficiency.

S3. It demonstrates the effectiveness of SSM in link prediction.

**Weaknesses:**

W1. The proposed task, “continuous predictions,” is somewhat confusing and may mislead readers into thinking it refers to predicting multiple continuous edges simultaneously. In fact, most existing TGNN methods already address what the authors describe as “continuous predictions.” For example, RNN-based approaches such as JODIE and TGN continuously adapt node memory states as the dynamic graph evolves, while attention-based methods are stateless and rely on recent interactions as input—making them inherently suitable for continuous prediction tasks. Given this, it is unclear why a new term, “continuous predictions,” needs to be introduced.

W2. Regarding efficiency, prior methods such as Instant have demonstrated superior performance compared to Coden. Moreover, attention-based models like DyGFormer do not require model updates as the dynamic graph evolves, thereby saving substantial update time. How do you calculate the training time of such methods?

W3. I also noticed that the authors of Instant proposed a follow-up work, Decoupled [R1], which further improves performance by integrating several sequential modeling strategies. The idea behind Coden appears conceptually similar, combining efficient updates with strong sequential modeling capabilities. Could the authors elaborate on the key improvements of Coden compared to Decoupled, beyond the introduction of the SSM for sequence modeling?
[R1] Zheng, Yanping, Zhewei Wei, and Jiajun Liu. "Decoupled graph neural networks for large dynamic graphs."

**Questions:**

See weaknesses.

---

### Official Review · Reviewer_RNiF · 2025-10-31

**Soundness:** 2
**Presentation:** 1
**Contribution:** 2
**Rating:** 2
**Confidence:** 3

**Summary:**

The authors propose a new Temporal Graph Neural Network architecture to tackle predictions in dynamic graphs at a number of discrete points. The method leverages a personalized page-rank approach for spatial mixing of node embeddings and a linear SSM for temporal mixing.

Experiments are carried out on a new setup requiring predictions at a sequence of discrete points in time for 5 commonly used dynamic graph datasets, showing that the proposed method beats other baselines.

**Strengths:**

The overall approach seems reminiscent of the [T-GCN] approach with a spatial component followed by a temporal one. The authors leverage previous personalized paged rank type of approaches like [DynAnom] for the spatial component with a simple linear state-space model for the temporal one. While individual components are not necessarily novel, their combination to achieve quasi-continuous predictions on dynamic graphs seems to be.

The authors also make an effort to evaluate many baselines on many datasets, from small to large scale ones like Papers100M. The results are well presented with average and best accuracies (over time) and using 10 trials per run.

[T-GCN] Zhao et al. T-GCN: A Temporal Graph ConvolutionalNetwork for Traffic Prediction
[DynAnom] Subset Node Anomaly Tracking over Large Dynamic Graphs

**Weaknesses:**

### 1. Poor Writing and Clarity
The paper is very poorly written and lacks clarity. In particular section 3, describing the proposed method, is impossible to understand without the help of the appendices. The authors should first explain their method at a high level, justifying their choices instead of jumping into specific details of the algorithm without any context or properly explaining the notation (e.g., what is the difference between $Z$ and $H$ in Equation 2? How is the latter an approximation of the former and why? Which is actually computed by the algorithm?).
The flow of the whole paper could use work. The introduction spends too much space on a straw man argument against snapshot-based static GNN methods instead of focusing on more relevant TGNN approaches. The limitations of existing methods are vague and so is how the proposed method relates to these approaches and its advantages. This is not improved by the related work section which, again, is poorly structured and doesn't cover Random-Walk based methods like [CAW-N].

### 2. Method and Complexity Analysis

From what I was able to decipher, in large part with recourse to the appendices, the time span of the dataset is split into a sequence of smaller (disjoint) time intervals:
1. Embeddings for all nodes in the graph, H, are computed and "continuously" updated with a parameterless page rank type of approach. These updates are batched for efficiency, triggered by a condition on approximation guarantee, with multiple updates being possible per time interval (and at least 1).
2. For each update to H a separate set of node states, M, is updated using a linear SSM over H. This is essentially like running a (learnable) exponential smoothing over the H states over time.
The authors don't clearly motivate the reason for this two step approach, nor do they ablate removing the SSM component.

It is unclear to me what the authors mean with their notation of $m^{(t)}$ in Table 1 and 6. Is it the total number of edges in the graph at time $t$ or the number of edges in a batch of updates?
I believe the authors might be misunderstanding some of the RNN based methods like [JODIE] and [TGN] which only spatially mix states as they are involved in updates. Given an observed event $(u, v, t, e)$, these methods only update the internal state ($M(u)$ and $M(v)$ in the authors' notation). This update is coupled but only depends on the previous states of both $u$ and $v$. This is highly efficient, at least as far as inference goes, as an update (or batch of updates) does not depend on the overall size of the graph as the proposed method seems to...
TGN does introduce a further (optional) graph attention based module computed on top of the internal states, but this is used only to compute an output embedding for prediction (it is not stored as a state).

Finally, while the method is put forward as working in continuous time, it does not efficiently produce continuous outputs (like many other TGNN models naturally can). The whole point of the method is that updates are "batched" for efficiency. This should clearly be recognized as a limitation in the paper.

### Experiments

- The setup the authors used seems to be directly tailored to their proposed method which requires splitting time into a (small) discrete number of prediction points. The authors claim as justification that:
> Note that this setting is substantially different from a general experimental setting in TGNNs, which typically focuses only on the fully formed final graph and fails to reflect the evolving process of CTDG

which is not the case in my opinion. Many TGNN papers focus on link or node-prediction tasks that require predictions at each event.
- It is unclear to me what the setup for the baselines is. Are the updates also batched in the same way as the proposed method? Such batching would seem highly detrimental for a model like TGN which works best with small batches of updates.
- The results are not so impressive. Performance seems very similar across many methods, often within standard errors. It seems possible that small tweaks to the setup to be more favorable to other methods might change the ordering.


### Other points

The comparison with Li et al., 2024, which also explores SSMs for CTDG's, in line 220 is very vague. The point for novelty in the SSM component is not well made.

Line 123: evolvment -> evolution

### References
[CAW-N] Wang et al. Inductive Representation Learning in Temporal Networks via Causal Anonymous Walks
[TGN] Rossi et al. Temporal Graph Networks for Deep Learning on Dynamic Graphs
[JODIE] Kumar et al. Predicting Dynamic Embedding Trajectory in Temporal Interaction Networks

**Questions:**

- What is the meaning of $m^{(t)}$ in Table 1 and 6?
- What is the setup for training the baselines (like TGN)?
- Can the authors more clearly explain how their application of a linear SSM for CTDGs differs from prior work?
- Why a linear SSM and not some other form of nonlinear recursion? The former have more limited representational capacity but can be parallelized over sequence length during training. The authors don't seem to take advantage of prefix-sums here so I wonder why choose a linear recurrence for graphs?
- Why not run TGN without the optional attention-based embedding module if training and inference time efficiency is one of the main concerns being evaluated?

---

### Official Review · Reviewer_7MTQ · 2025-11-01

**Soundness:** 3
**Presentation:** 3
**Contribution:** 3
**Rating:** 8
**Confidence:** 3

**Summary:**

The paper addresses the problem of continuous prediction in temporal GNNs.
The work introdues CODEN, which is a framework that aims at balancing two competing aspects, namely he need to model long-term temporal dependencies, while limiting the computational resources.
This effect is obtained by a clever combination of some simple yet effective components, leveraging approximate embeddings (remarkably, coming with an accuracy guarantee), a lazy update of the state which is activate only when needed, an a time evolution based on state-space models.
The experiments convincingly demonstrate the effectiveness (accuracy and reduced complexity) of the framework.

**Strengths:**

- The paper is well written, and the presentation has a very good clarity and precision.
- The efficient approximation of the non-parametrix PPR-embedding is a clever idea. Especially, the error guarantees are the fundamental tool to make the lazy evaluation very principled.

**Weaknesses:**

**On the presentation of the approximate embedding:**
- The parameter $\alpha$ in (2) is never defined nor discussed. Are there indications to select it?
- The approximate embedding $H^{(t)}$ is one of the central parts of the work, but its definition is sometimes confusing, especially around two aspects: (i) Based on my understanding of Appendix A.2, Lemma 1 and Proposition 1 are valid for _any_ approximate embedding $H^{(t)}$ which satisfies $||H^{(t)} - Z^{(t)}||_1\leq\varepsilon n^{(t)}$. If this is the case, this is an interesting insight, which adds generality to the results and gives possible ground for extensions. So I suggest to mention this fact. (ii) On the other hand, the actual definition of $H^{(t)}$ is never discussed in the paper (except for a few hints in the introduction), as it is deferred to Appendix A.6.6 and A.8. A reader is left with the feeling of not knowing how this crucial component is defined. I suggest to give at least some summary of the main ideas in the main text.

**On the accuracy/efficiency tradeoff:**

CODEN effectively implements this tradeoff by the parameter $\varepsilon$, which is reasonably set to $\mathcal O(1/n^2)$. However, by steering it it should be possible to move from a very fine-grained, costly updating scheme, to a more coarse but efficient one. I believe this is an important aspect and possibly an added value of CODEN, but it is not sufficiently discussed in the paper nor tested in the experiments.

**Questions:**

Apart from the points discussed above, there are the following minor points:

- The variable T is used to denote both the total time (see e.g. first line below Table 1) and the set $\\{t_1, t_2, \dots\\}$ ("Problem definition" on page 3).

- Right above (1): the sentence "given a new edge connecting $e_{t+1} = {u, v, t + 1}$ from node u to node v" is unclear. Perhaps "given a new edge $e_{t+1} = \\{u, v, t + 1\\}$ connecting node u to node v" would be better.

- The sentence "mentioned above, we effectively control the extent [...] essential temporal dependencies within the graph" is unclear.

- I could not locate the proof of Proposition 3.

---

### Official Review · Reviewer_x495 · 2025-11-03

**Soundness:** 3
**Presentation:** 3
**Contribution:** 2
**Rating:** 4
**Confidence:** 5

**Summary:**

This paper studies the problem of continuous prediction in dynamic graph learning, where models must efficiently update predictions as the graph evolves over time. The authors propose CODEN, a state-space–based temporal GNN that performs incremental embedding updates with bounded error and introduces a lazy-sampling mechanism to improve efficiency.

**Strengths:**

1. The problem of continuous prediction in dynamic graph learning is important and practical.

2. The authors propose an efficient incremental updating mechanism with theoretical error guarantees.

3. The experimental results demonstrate strong scalability, especially on large-scale datasets such as Papers100M.

**Weaknesses:**

1. The paper’s novelty appears limited. The proposed continuous prediction setting is conceptually very similar to the live-update evaluation in ROLAND, while the theoretical analysis on incremental computation and bounded errors closely follows prior work such as Instant and IDOL. Moreover, the proposed lazy sampling threshold essentially mirrors the adaptive training mechanism introduced in Instant.

2. While the empirical results are impressive, I am concerned about the fairness of the experimental setup. The paper positions CODEN as a framework for continuous prediction, i.e., continuously updating predictions for the current state as the graph evolves. However, several baselines—such as EvolveGCN, TGN, and DyGFormer—were originally designed for future prediction tasks, where the goal is to infer future states from historical information. Forcing these models to operate in a continuous-prediction regime (by requiring them to re-run updates and output current predictions at each time step) may disadvantage them both computationally and conceptually. Moreover, ROLAND and other recent DTDG methods should be included as baselines.

3. Although the paper claims that the proposed SSM-based temporal modeling improves over RNN-based TGNNs such as TGN and EvolveGCN, no direct empirical comparison is provided. The ablation study only contrasts SSM with attention and static variants, but not with an RNN-based version. It remains unclear whether the observed efficiency and stability gains stem from the SSM design itself or simply from other architectural factors. Including a direct RNN–SSM comparison in terms of accuracy, runtime, and parameter count would significantly strengthen the paper’s claims.

**Questions:**

See Weaknesses

---

### Meta-Review · Area_Chair_H1Xc · 2026-01-01

**Summary:**

The paper introduces CODEN, a temporal GNN designed for continuous prediction on evolving graphs by combining fast incremental node-embedding updates with bounded approximation error and a state-space model (SSM) to capture temporal dynamics. The approach aims to preserve model accuracy while reducing computation as the graph changes. The authors provide theoretical efficiency guarantees for the incremental scheme and evaluate on several dynamic-graph datasets.

The authors did not participate in the rebuttal. Reviewer concerns were not resolved.

**Reviewer Concerns:**

The authors did not participate in the rebuttal. The key reviewer concerns of the paper include:

1/ The paper’s novelty is limited. The proposed continuous prediction setting is not new. It is similar to live-update evaluation in ROLAND (x495) and most existing TGNN methods, e.g.,  JODIE and TGN, already address it (uUcu).

2/ The fairness of the experiments are challenged. The evaluation settings favor CODEN. (x495, RNiF).

3/ The point for novelty in the SSM component is not well-made. (x495, RNiF, uUcu)

**Reviewer Scores:**

Three reviewers gave negative scores (two gave 4 and one gave 2. One reviewer gave a positive score, i.e., 8. There are no rebuttal discussions.

---

### Decision · Program_Chairs · 2026-01-26

Reject